# A yeast surface display platform for characterizing CAR T cell responses to cancer antigens

Marcus Deichmann [1], Giovanni Schiesaro [1], Keerthana Ramanathan [2], Katrine Zeeberg[1], Nanna M. T. Koefoed[1], Maria Ormhøj[2], Rasmus U. W. Friis[2], Ryan T. Gill[1,3], Sine R. Hadrup [2], Emil D. Jensen [1] ✉ & Michael K. Jensen [1] ✉

Chimeric antigen receptor (CAR) T cells have become an established immunotherapy with promising results for the treatment of hematological malignancies. However, modulation of the targeted antigen's surface level in cancer cells affects the quality and safety of CAR-T cell therapy. Here we present an engineered yeast-based antigen system for simulation of cancer cells with precise regulation of surface-antigen densities, providing a tool for controlled activation of CAR T cells and systematic assessment of antigen density effects. This Synthetic Cellular Advanced Signal Adapter (SCASA) system uses G protein-coupled receptor signaling to control cancer antigen densities on the yeast surface and provides a customizable platform allowing selectable signal inputs and modular pathway engineering for precise output fine-tuning. In relation to CD19+ cancers, we demonstrate synthetic cellular communication between CD19-displaying yeast and human CAR T cells as well as applications in high-throughput characterization of different CAR designs. We show that yeast is an alternative to conventional technologies (e.g. microbeads) and can provide higher activation control of clinically derived CAR T cells in vitro, relative to cancer cells. In summary, we present a customizable yeast-based platform for high-throughput characterization of CAR-T cell functionality and show potential applications within therapeutic T cells in clinical settings.

Over the past two decades, synthetic biology has enabled new generations of cell therapies through novel genetic engineering strategies and programmable gene circuits to achieve controlled and therapeutically balanced functions in both microbes and human cells[1]. Cellular immunotherapies show promising clinical results for the treatment of cancers, notably with chimeric antigen receptor (CAR) technology, which commonly relies on autologous T cells engineered to heterologously express a synthetic CAR construct enabling T cell anticancer responses targeted towards cancer-associated antigens (e.g. CD19), upon patient re-infusion[2,3]. Since 2017, seven FDA

approvals on CAR products for hematological cancers have been registered[3,4], including anti-CD19 CAR T cells for relapsed or refractory (r/r) B-cell acute lymphoblastic leukemia (B-ALL), showing complete remission (CR) in 70–96% of patients[5–14].

However, while remission rates are encouraging, 14–57% of r/r B-ALL patients obtaining CR from CAR T cell therapy eventually experience relapse[5–14]. Some relapses are caused by immunosuppression, exhaustion, or poorly persistent and ineffectual CAR T cells[14–17], and others are attributed directly to escape via antigen modulation by cancer cells, where CAR T cells may remain functional, but simply lack

[1]Novo Nordisk Foundation Center for Biosustainability, Technical University of Denmark, Kongens Lyngby, Denmark. [2]Department of Health Technology, Technical University of Denmark, Kongens Lyngby, Denmark. [3]Artisan Bio, Louisville, CO, USA. ✉e-mail: emdaje@biosustain.dtu.dk; michaelkroghjensen@gmail.com

sufficient targetable antigen[18]. CD19 modulation occurs through a variety of mechanisms[18–26] and is thought to be the primary mechanism of tumor escape in r/r B-ALL, occurring in 7–28% of r/r B-ALL CRs, is a major determinant for response durability, and is associated with poor prognosis[3,5–10,13,14,27–30]. Similar challenges are also observed in other cancers, such as diffuse large B-cell lymphoma (DLBCL)[3,18,31,32].

Even gradual modulation that causes decrease in the cancer-cell-surface antigen density can increase resistance to therapy, as CAR T cell responses are graded to antigen density[26,33–42]. CAR T cell efficiency and behavior are directly influenced, as distinct anticancer responses are triggered at different antigen-density thresholds and have density-dependent intensities. For example, cytolytic activity requires a lower antigen density than CAR T cell proliferation and cytokine release[33–36,38,39]. These antigen density-dependent effects have differential impact across CAR designs and CAR expression levels[26,33,34,37,38,40,41], emphasizing the importance of strategic CAR design to mitigate antigen modulation, as well as proper characterization of the hundreds of novel CARs for new antigens and cancers undergoing development and clinical trials[43,44]. Antigen-density model systems continue to improve the understanding of these dynamics, such as cell lines with varying antigen densities that closely mimic the complexity of in vivo cell-to-cell interactions[26,33–40,45–47], as well as non-cellular materials embedded with antigens that enable greater antigen-density control and orthogonality[34,41,42,48–52]. Such preclinical models remain crucial to the advancement of CAR T cell therapy[53]. Complementing these approaches, engineered yeast cells can offer a highly versatile alternative.

The concept of using engineered yeast cells (*Saccharomyces cerevisiae*) as T-cell response screening platforms through co-cultivation emerged in literature in 1998 with the advent of yeast-surface display (YSD) technology[54,55]. Since then, yeast has become an applied tool in deciphering T-cell receptor (TCR) specificities and peptide-loaded major histocompatibility complexes (pMHC)[56–58], and co-cultivation has been employed to verify pMHC-display functionality and directly assay TCR-specific T-cell activation[59–64]. In this context, yeast platforms benefit from a well-developed genetic engineering toolbox and an extensive repertoire of gene-circuit parts for efficient and versatile customizability. Furthermore, DNA-encoding in yeast supports high-throughput interaction screening of libraries and directed evolution of human proteins. More practically, the availability and simplicity of implementation, as well as

robustness and fast growth as a self-replicating system, makes yeast a powerful and cost-effective platform for synthetic interspecies cell-cell communication[56,60,63,65,66].

In this study, we leverage these traits to develop a controllable antigen-density model system, and build cancer-simulating CD19+ yeast to study the activation dynamics of human CAR T cells ex vivo. To achieve this goal, we create a fully genome-encoded cellular platform, named the Synthetic Cellular Advanced Signal Adapter (SCASA) system, which can be used to control surface-displayed CD19 at levels spanning >3 orders of magnitude and is compatible with T cell co-cultivation with no inherent effect on viability, proliferation, or activation of donor-derived T cells or cell lines. This allows dynamic control of CAR-specific human immune cell activation at levels on par with the standard NALM6 cancer cell line. Further, we apply SCASA yeast cells to characterize and discriminate antigen-dependent effects on signaling pathways in different CAR designs with comparisons to commonly applied technologies. Lastly, we functionally assess a donor-derived CAR T cell product using yeast, with an activation efficiency and robustness in antigen density and target cell numbers beyond what a cancer cell line can support. With these findings, we position yeast as a promising, accessible synthetic biology platform for effective and customizable assays in cell therapy, enabling pre-clinical screening of candidate designs, detailed characterization of therapeutic responses, and batch verification of clinical products.

## Results

### Designing the Synthetic Cellular Advanced Signal Adapter (SCASA) system

In the interest of establishing a versatile platform technology for evaluating antigen-density effects in immune cell responses, we envisioned the SCASA system comprising yeast-based artificial antigen-presenting cells (aAPC). The system allows yeast to surface-display proteins (*the effector module*) at levels regulated by an engineered version of the well-characterized and highly modular yeast pheromone response pathway (PRP)[67–71] (*the processing module*), from a sensed amount of a chosen extracellular G protein-coupled receptor (GPCR) ligand presented to the cells (*the sensory module*) (Fig. 1a). Ideally, such a design comprises controllability of display density, tunability of signal processing dynamics, orthogonality to T cell activation, and customizability in the types and intensity of

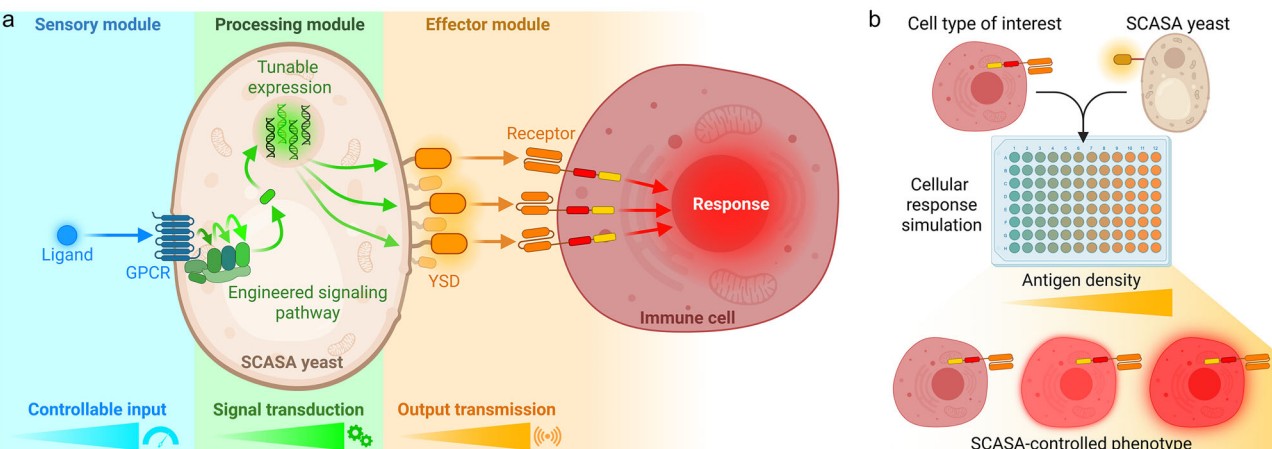

**Fig. 1 | Outline of the Synthetic Cellular Advanced Signal Adapter (SCASA) system. a** SCASA yeast cells engineered to interact with human immune cells. The yeast can present tunable dosages of signal molecules (e.g. antigens) via yeast surface display (YSD) on the cell surface (*effector module*), proportional to the intracellular signal transduced via an engineered pheromone response pathway (*processing module*) with controlled activation and intensity by G protein-coupled

receptors (GPCR) based on cognate ligands presented to the cells (*sensory module*). Each module is customizable to adapt to the specific application of the SCASA yeast cells. **b** Example of a co-cultivation layout in which engineered SCASA yeast cells are used to simulate different antigen densities of a chimeric antigen receptor (CAR) T cell target to obtain gradual intensities of CAR T cell responses. Created in BioRender. Deichmann, M. (2025): https://BioRender.com/dgx7343.

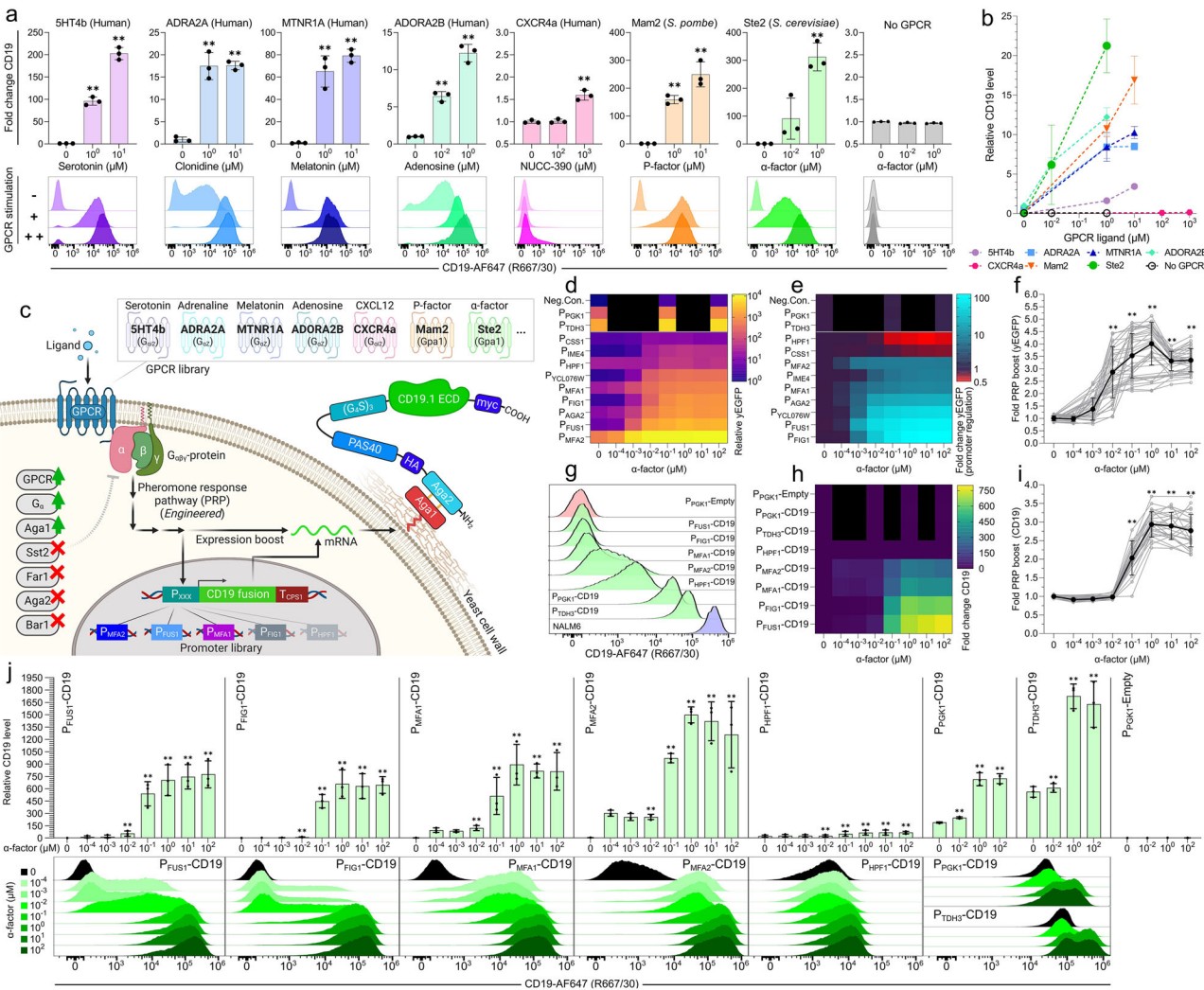

**Fig. 2 | Characterization of the SCASA system for CD19 antigen presentation by yeast cells. a** Heterologous G protein-coupled receptors (GPCR) as sensory modules for controlling CD19 display with cognate agonists: fold change CD19 levels (*top row*) and absolute CD19 histograms (*lower row*). All strains employ $P_{FUS1}$ and vary between GPCRs and $G_\alpha$-subunits. **b** Comparison of CD19 levels across GPCRs. **c** CD19 SCASA yeast design; CD19 expression is controllable by GPCR-dependent ligands (*sensory module*). System regulation depends on signaling through an engineered pheromone response pathway (PRP) via $G_{\alpha\beta\gamma}$-protein, choice of promoter, and expression boost effects (*processing module*). CD19 output is a fusion protein composed of HA- and myc-tags, PAS40-linker, $(G_4S)_3$-linker, and CD19.1 ECD, fused to Aga2 (*effector module*). Strains were optimized by; $G_\alpha$-subunit, GPCR, and Aga1 overexpression, as well as gene knock-outs; *ste2Δ0, ste3Δ0, gpa1Δ0, sst2Δ0, bar1Δ0, far1Δ0, aga2Δ0*. Created in BioRender. Deichmann, M. (2025): https://BioRender.com/dtuelpw. **d** Relative yEGFP levels of promoters with α-factor stimulation including PRP boost effects, sorted; low to high (5 h. post-induction). **e** Fold change promoter induction (yEGFP), excluding PRP boost effects (SSC-

normalization), sorted; low to high. **f** Approximated PRP boost of yEGFP expression from quantified PRP activation (*black*) across all strains and designs (*grey*) (*n* = 27). **g** CD19 histograms of SCASA yeast with different promoters without GPCR stimulation (*green*), a $P_{PGK1}$-Empty control lacking CD19 (*red*), and NALM6 (*blue*). **h** Fold change CD19 of SCASA yeast during GPCR stimulation with α-factor (20 h. post-induction). **i** Approximated PRP boost of CD19 display from quantified PRP activation (*black*), across all strains and designs (*grey*) (*n* = 15). **j** Comparison of CD19 levels during GPCR stimulation of SCASA yeast, relative to lowest detected CD19 level (*top row*), and CD19 histograms (*lower row*). Unless otherwise noted, data is means of median fluorescence intensities (mMFI) for biological replicates (*n* = 3) and standard deviations hereof. Histograms are representative replicates normalized to the mode. Statistical tests: One- and Two-way ANOVA with multiple comparisons statistical tests. Significance levels: \**p* ≤ 0.05, \*\**p* ≤ 0.001. Not all pairwise comparisons are shown. All statistics and extended analyses in: Supplementary Figs. 1–12 and Supplementary Data 1–6. Source data provided as a Source Data file.

sense-response functions, which would be favorable for adapting the SCASA system to user-defined cell assays and health applications[1,56] (Fig. 1b).

### CD19 yeast surface display for cancer cell simulation

In our first design for antigen display, we employed a mutated CD19 extracellular domain (CD19.1 ECD) in an Aga2-fusion display construct for correct folding and expression in yeast[72,73], hereafter merely denoted CD19 (Fig. 2). Functional CD19 display was confirmed via staining with FMC63-clone anti-CD19 antibody containing the binding domain of current FDA-approved CAR T cell designs[3]. YSD was

optimized by overexpression of the Aga1 anchor and deletion of native Aga2 (*aga2Δ0*) to avoid competitive binding to Aga1. To both increase antigen-display efficiency and limit inconsistent behavior in antigen-displaying populations, genetic display cassettes were genomically integrated rather than expressed from a plasmid[63,74–76]. Ultimately, this yielded 99.7 ± 0.3% CD19+ yeast cells, clearly exemplified by the $P_{TDH3}$ promoter, thus closely resembling the phenotype of the cancer benchmark CD19+ NALM6 B-cell leukemia cell line (100 ± 0% CD19+) (Fig. 2, Supplementary Fig. 1). We additionally confirmed the increased efficiency of genome integration on galactose-induced CD19 expression from $P_{GAL1}$ (Supplementary Fig. 2), though this traditional system

was not further pursued due to its limitation to a single input and limited customizability[55,77–79].

## Sensory modules using heterologous GPCRs allow for customizable input

To empower SCASA yeast cells with versatile and customizable input-sensing capabilities, we next assessed if the sensory module could comprise GPCRs coupled to an engineered yeast PRP, which normally mediates signaling during yeast mating[68,69,71] (Fig. 2a–c). The extracellular sensing of GPCRs displays exceptional diversity in the types of specific ligands, ranging from light rays and small molecules to peptides and large complex proteins[67–71,80,81]. Accordingly, the constructed strain library encoded individual heterologously expressed GPCRs for sensing a diverse set of ligands, with CD19 expression controlled by the PRP-regulated promoter $P_{FUS1}$[82] (Fig. 2c). Upon administration of GPCR agonists to the SCASA yeast cells, all designs showed controlled functional up-regulation of CD19 levels ($p < 0.0001$) (Fig. 2a, Supplementary Fig. 3). In the tested concentration ranges, the highest absolute CD19 levels and fold changes were provided by fungal peptide-sensing GPCRs Ste2 (α-factor) at $312.5 \pm 50.0$-fold and Mam2[68] (P-factor) at $249.4 \pm 44.9$-fold (Fig. 2a-b), while human GPCRs sensing small-molecule neurotransmitters showed lower absolute CD19 levels (Fig. 2b) and fold changes, with 5HT4b[83] (serotonin) at $202.6 \pm 13.8$-fold, MTNR1A[68] (melatonin) at $79.2 \pm 5.9$-fold, ADRA2A[69] (adrenaline) at $17.6 \pm 0.9$-fold, and ADORA2B[68] (adenosine) at $12.2 \pm 1.2$-fold (Fig. 2a). Lastly, the human chemokine-sensing CXCR4a[84] (CXCL12/SDF1) showed a modest $1.6 \pm 0.1$-fold up-regulation of CD19 levels (Fig. 2a). The GPCRs caused significantly different unstimulated background CD19 levels, notably with ADORA2B causing a $14.0 \pm 0.8$-fold background increase ($p \leq 0.0013$) and 5HT4b lowering the background to $0.24 \pm 0.02$-fold the level of cells lacking GPCRs ($p < 0.0001$) (Fig. 2a).

In conclusion, the sensory module was customizable, as exemplified by the individual expression of six heterologous GPCRs enabling the adoption of small molecule-, peptide-, and protein-based input signals, and providing a 1,249-fold span in surface-displayed CD19 levels across the examined range of GPCR stimulation.

## An engineered pheromone response pathway enables a customizable processing module

As CAR T cell products show different sensitivities to changes in antigen densities[26,33,34,37,38,40,41], a high degree of adaptability of SCASA cell designs is advantageous to enable simulation of the relevant ranges of antigen densities for a specific CAR design. In this regard, PRP-coupled GPCR signaling in yeast has several valuable attributes, such as highly sensitive concentration-dependent analog signal transduction and a large dynamic output range, combined with its engineerability characterized by high modularity and tunability[67–69,71] (Fig. 2). To generate well-characterized processing module parts for tuning surface display levels, we characterized 11 candidate promoters through yEGFP-expression (Fig. 2d, e). The promoters were selected from our recent transcriptome analysis of GPCR-mediated PRP activation[67], showing expression ranging from undetectable mRNA levels to the most highly expressed gene in *S. cerevisiae*, and spanning 6 orders of magnitude in mRNA abundance upon GPCR stimulation (Supplementary Fig. 4, Supplementary Data 1). The processing module promoter library spanned a 1025-fold range in basal yEGFP output intensity (Fig. 2d, Supplementary Fig. 5), and following ligand-mediated GPCR stimulation, the individual promoters displayed further up- or down-regulated dynamics (Fig. 2e, Supplementary Fig. 6), providing a 6866-fold span in total library output range (Fig. 2d). Characterization of yEGFP revealed that output dynamics were highly dependent on the choice of promoter, emphasized by response variation being attributed mainly to promoter choice (SV = 75.2%), rather than GPCR stimulation level (SV = 13.4%) (Fig. 2d, e, Supplementary Data 2). Output

was boosted by an overall post-transcriptional promoter-independent multiplicative amplification of expression, also observed in similar systems[68,85], at an intensity dependent on the degree of yeast mating phenotype induction ($p < 0.0001$) (Fig. 2f). This amplification coincided with shmooing, a morphological key indicator of PRP activation[67] (Supplementary Fig. 7), and was dependent on the main yeast mating response transcription factor Ste12[86] ($p < 0.0001$) (Supplementary Fig. 8). In this case, yEGFP expression was generally boosted by up to $4.0 \pm 0.9$-fold (Fig. 2f), exemplified by the regulated output for $P_{PGK1}$ and $P_{TDH3}$ in response to GPCR stimulation (Fig. 2d, Supplementary Fig. 9), despite these genes showing no up-regulation of mRNA levels (Supplementary Fig. 4, Supplementary Data 1), and commonly being regarded as constitutive promoters[67,87].

Based on these learnings, we generated a yeast strain library to simulate the antigen heterogeneity of CD19+ cancer cells through YSD, with antigen variations enabled by promoter choice and GPCR stimulation (Fig. 2g–j). Baseline constitutive expression levels of the seven chosen promoters yielded a 562-fold span of CD19 levels across the strain library, with the promoter $P_{TDH3}$ reaching a CD19 level of $20.3 \pm 2.9\%$ of NALM6 (Fig. 2g, Supplementary Fig. 10). For all strains, CD19 levels could be significantly diversified from their baseline through GPCR stimulation (Ste2) by α-factor (0-100 μM) resulting in a multitude of different CD19+ profiles ($p < 0.0001$) (Fig. 2j). Specifically, the highest CD19 levels were obtained for $P_{TDH3}$ at $1,724 \pm 151$-fold higher than the lowest basal CD19+ level ($p \leq 0.0001$) (Fig. 2j). However, $P_{MFA2}$ showed the largest span in absolute CD19 levels upon stimulation ($p \leq 0.0361$) (Supplementary Fig. 11), by increasing $322.7 \pm 20.7$-fold (Fig. 2h, Supplementary Fig. 12), to a level that was $2.7 \pm 0.4$-fold higher than baseline $P_{TDH3}$ expression, commonly regarded as the strongest constitutive yeast promoter[87] (Fig. 2j). $P_{FUS1}$, $P_{FIG1}$, $P_{MFA1}$, and $P_{MFA2}$ could all provide drastic changes in CD19 levels relative to the unstimulated condition ($p < 0.0001$) (Fig. 2j), with $P_{FUS1}$ ($774 \pm 165$-fold) and $P_{FIG1}$ ($630 \pm 99$-fold) providing up-regulation from undetectable CD19 to levels comparable to $P_{TDH3}$, highlighting that these individual strains were capable of spanning the entire CD19 range of the unstimulated strain library (Fig. 2g–j). Surface display levels were also boosted by PRP activation, providing an estimated maximum PRP boost effect of $2.9 \pm 0.3$-fold increase in CD19 ($p < 0.0001$) (Fig. 2i), which was attributed to GPCR stimulation (SV = 94.7%) and was equivalent across strain design differences (SV = 0.9%) (Supplementary Data 5). Remarkably, the change in absolute output from the $3.1 \pm 0.3$-fold up-regulation of $P_{TDH3}$, resulting from the PRP boost, was greater than the $774 \pm 165$-fold up-regulation of $P_{FUS1}$ ($150.2 \pm 37.5\%$, $p = 0.012$) (Supplementary Fig. 11), hence the PRP boost allows high-intensity antigen presentation and larger diversity in the library.

In summary, CD19 levels were controllable on SCASA yeast cells, enabling CD19+ profiles spanning up to 1,724-fold in intensity. Importantly, the control of CD19 levels was I) dependent on the GPCR ligand concentration, II) showing no residual non-displaying populations, III) specific for the chosen processing module promoter, and IV) relying on two separate effects, namely promoter regulation and a post-transcriptional PRP boost caused by phenotypic changes to the yeast cell (Fig. 2).

## Co-cultivation of yeast and human T cells

Before proceeding to yeast-mediated CAR T cell activation assays, we first assessed physiological properties of the co-cultivation of yeast with human immune cells. To confirm that undesired yeast growth would not occur, we first established that yeast showed negligible growth in T cell media, RPMI + 10%FBS and ImmunoCult™-XF T Cell Expansion Medium, not initiating growth for at least 22 h., providing stable yeast cell numbers during co-cultivation (Supplementary Fig. 13). Secondly, to assess T cell viability and proliferation in the presence of yeast, we co-cultivated yeast with donor-derived naïve

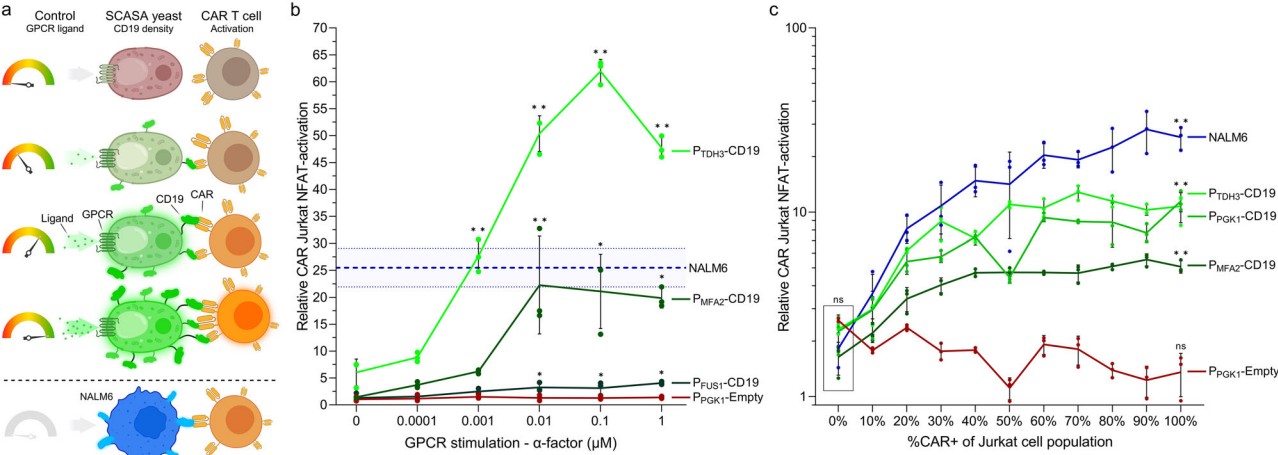

**Fig. 3 | Activation of 4-1BB CAR Jurkat NFAT-Luc cells using SCASA yeast cells.** **a** Illustration of chimeric antigen receptor (CAR) Jurkat cell co-culture with SCASA yeast cells. Jurkat cells that express anti-CD19 CARs (*CAR+* ) bind to CD19+ SCASA yeast cells, analogously to as for the CD19+ NALM6 human cancer cell line. Here, Jurkat cells contain a NFAT-Luc reporter system, which upon activation expresses luciferase. SCASA yeast cells can be stimulated through their G protein-coupled receptors (GPCR) to induce differential CD19 antigen density (e.g. Ste2 with ligand α-factor). Created in BioRender. Deichmann, M. (2025): https://BioRender.com/urlsdg9. **b** Relative CAR activation in co-cultures of 100% CAR+ Jurkat NFAT-Luc cells with SCASA yeast designs; $P_{FUS1}$-CD19, $P_{MFA2}$-CD19, and $P_{TDH3}$-CD19 (*green lines*) at a target-to-effector (T/E) cell ratio of 0.2x yeast cell per Jurkat cell for 18 h., with increasing SCASA yeast GPCR stimulation for CD19 display control (α-factor). Negative control: $P_{PGK1}$-Empty lacking the CD19 CDS in the display construct (*red line*). Positive benchmark control: NALM6 (*blue line*). One-way ANOVA with

Dunnett's multiple comparisons statistical test of NFAT-activation compared to 0 μM is shown for each strain. **c** Relative CAR activation ($\log_{10}$-scale) in co-cultures of Jurkat NFAT-Luc cells with unstimulated SCASA yeast designs; $P_{MFA2}$-CD19, $P_{PGK1}$-CD19, and $P_{TDH3}$-CD19, as well as a negative control $P_{PGK1}$-Empty, and NALM6 as benchmark control. The percentage of Jurkat NFAT-Luc cells expressing CARs was graded from 0–100% CAR+ . Two-way ANOVAs are shown for a comparison of background activation at 0% CAR+ (Tukey's multiple comparisons test) (*box*), and significant activation within the 0–100% CAR+ range for each SCASA yeast strain and NALM6 relative to 0% CAR+ (Dunnett's multiple comparisons test). Data is based on means of three biological replicates ($n = 3$) and standard deviations hereof. Significance levels: **ns**: not significant, *$p \leq 0.05$, **$p \leq 0.001$. Not all pairwise comparisons are shown. All statistics and extended analyses in: Supplementary Fig. 14 and Supplementary Data 8. Source data provided as a Source Data file.

T cells at cellular ratios of 0.5x, 1.0x, and 10.0x yeast cells with compliance to T cell growth requirements for 96 h. Here we found that T cell viability and proliferation were not affected by the presence of yeast at any ratio (Supplementary Fig. 13). Lastly, we verified that the SCASA system of the engineered yeast cells remained functional during T cell growth conditions, despite the constrained growth in T cell media (Supplementary Fig. 13).

In summary, all experiments indicated that co-cultivation of yeast and T cells was possible under standard T cell growth requirements without affecting the viability or inducing proliferation in donor-derived naïve T cells, and with yeast cells showing constrained growth yet retaining SCASA system functionality.

**Controlled human immune cell activation by SCASA yeast cells**
We proceeded to investigate the ability of SCASA yeast cells to confer communication with immune cells of relevance to immuno-oncological applications, specifically, CAR T cells. Here, a Jurkat cell line incorporating a luciferase reporter system linked to the T cell activation transcription factor NFAT (Jurkat NFAT-Luc) was transduced with lentiviral vectors to express the FDA-approved anti-CD19 CAR, FMC63-CD8α-4-1BB-CD3ζ (4-1BB CAR) used in tisagenlecleucel[3,44,88]. Consequently, CAR T cells provided a bioluminescent signal of proportional intensity to activation imposed by target CD19+ yeast cells (Fig. 3).

To demonstrate controllable synthetic cell-to-cell communication, we diversified the CD19 levels on target SCASA yeast cells by GPCR stimulation, to simulate cancer cells with varying CD19 antigen densities, and performed co-cultivations with CAR Jurkat NFAT-Luc cells at a low target-to-effector cell ratio of 0.2x (Fig. 3a). Upon GPCR stimulation, all CD19+ yeast designs significantly induced activation of 4-1BB CAR Jurkat cells ($p \leq 0.0045$), with graded NFAT responses dependent on both the GPCR stimulation level and processing module promoter ($p < 0.0001$) (Fig. 3b). Hence, SCASA yeast activated CAR

Jurkat cells in an antigen density-dependent manner. The most intense CAR Jurkat cell activation was enabled by the $P_{TDH3}$-CD19 yeast design, yielding $61.9 \pm 2.2$-fold increased NFAT activity relative to a control monoculture ($p < 0.0001$), which exceeded the activation elicited by NALM6 by $2.4 \pm 0.4$-fold ($p < 0.0001$) (Fig. 3b).

To confirm the specificity of SCASA yeast-induced CAR activation, we made co-cultivations with a constant number of Jurkat cells that varied in the proportion of CAR+ cells, spanning from 0-100% CAR+ (Fig. 3c). Yeast containing CD19 in the display design could all significantly activate CAR Jurkat cells ($p < 0.0001$) (Fig. 3b, c), while a CD19[neg] control yeast ($P_{PGK1}$-Empty) never induced activation under any conditions of 0–100% CAR+ levels (Fig. 3c) or GPCR stimulation levels (Fig. 3b). Likewise, no activated Jurkat cells were detected in the range of 0–10% CAR+ for any co-cultivation, nevertheless, any >10% CAR+ increase provided significant responses for NALM6 and CD19+ yeast designs only ($p \leq 0.0331$). Hence, CAR Jurkat cell activation relied entirely on CD19 presentation by yeast cells and CAR expression in Jurkat cells (Fig. 3b, c). Given the lack of indications that yeast itself can activate T cells, at least via NFAT, these results support the idea of yeast cells as an orthogonal platform for assaying CAR T cell activation.

Each CD19+ yeast design provided significantly different activation dynamics in the 0–100% CAR+ range ($p < 0.0001$) (Fig. 3c) and across GPCR stimulation ($p < 0.0001$) (Fig. 3b), corresponding to the individual yeast CD19 levels (Fig. 2j). Responses showed that 4-1BB CAR-mediated NFAT activation was differentially sensitive to antigen densities. Specifically, high basal CD19 levels provided higher dynamic range and sensitivity to GPCR stimulation (i.e. $P_{MFA2}$-CD19 and $P_{TDH3}$-CD19), indicating that antigen densities satisfied the effective activation threshold (Fig. 3b, Supplementary Fig. 14). Accordingly, insufficient absolute CD19 to reach threshold levels for effective activation explains why the ~775-fold CD19 range of $P_{FUS1}$-CD19 (Fig. 2h) was not reflected in the CAR Jurkat cell response at this target-to-effector ratio (Fig. 3b), which only increased by $3.1 \pm 0.2$-fold (Supplementary Fig. 14).

In summary, CAR T cells (Jurkat NFAT-Luc) were activated by CD19-presenting SCASA yeast cells in a manner that was I) CD19-specific, II) CAR-specific, III) sensitive to CD19 antigen densities in regards to the intensity of NFAT activation, IV) different for each SCASA yeast cell design, V) controllable through GPCR stimulation of yeast cells, VI) capable of surpassing activation levels of NALM6, and VII) unaffected by the presence of yeast itself (Fig. 3b, c).

## SCASA yeast applications in characterizing CAR designs and downstream signaling

We next sought to demonstrate the direct application of CD19+ SCASA yeast cells in characterizing responses of different CAR designs, employing 50 distinct yeast-based stimulatory conditions of antigen densities and target-to-effector cell ratios to simulate different cancer scenarios. We chose to compare the two most commonly used FDA-approved designs, namely the 4-1BB CAR (FMC63-CD8α-4-1BB-CD3ζ) and the CAR used in axicabtagene ciloleucel and brexucabtagene autoleucel, FMC63-CD28-CD28-CD3ζ[3,44], referred to as the CD28 CAR. These CARs contain different hinge, transmembrane, and co-stimulatory domains, which have previously been shown to cause significantly different responses to antigen-density variations[26,38,44,47]. To quantify activation, we used a triple-parameter-reporter Jurkat T-cell line (TPR Jurkat cells) providing fluorescent output for three crucial T-cell activation transcription factors; NF-κB, NFAT, and AP-1, involved in proliferation, differentiation, and effector functions[44,89] (Fig. 4a).

CAR signaling significantly increased in response to co-cultivation with CD19+ SCASA yeast for NF-κB ($\leq 21.6 \pm 0.9$-fold), NFAT ($\leq 8.1 \pm 1.1$-fold), and AP-1 ($\leq 2.0 \pm 0.1$-fold) ($p < 0.0001$) (Fig. 4b). Systematic variations in response dynamics revealed CAR-specific signaling between CD28 and 4-1BB CARs, and distinct NF-κB, NFAT, and AP-1 signaling, which all were differentially dependent on antigen densities (Fig. 4b, Supplementary Figs. 15–17). CD28 and 4-1BB CAR designs notably affected NF-κB and NFAT, whereas AP-1 was more consistent between the designs across all conditions ($\leq 1.5 \pm 0.1$-fold) (Supplementary Fig. 15). Specifically, the CD28 CAR caused higher baseline NF-κB activity than the 4-1BB CAR independently of CAR stimulation ($2.6 \pm 0.1$-fold, $p < 0.0001$) and stronger NF-κB signaling upon CAR stimulation ($\leq 5.6 \pm 0.1$-fold) (Supplementary Fig. 15). NFAT was affected differently, as the CD28 and 4-1BB CAR had similar NFAT baselines, but upon antigen-density increase, NFAT signaling was more intensely activated by the CD28 CAR than the 4-1BB CAR ($\leq 2.9 \pm 0.4$-fold, $p < 0.0001$) (Supplementary Fig. 15). We performed regressions to obtain responsiveness coefficients that confirmed differential sensitivity to changes in antigen density and target cell ratios for both CARs ($p < 0.0001$) (Supplementary Figs. 18-19). The CD28 CAR responded with higher dynamicity than the 4-1BB CAR ($\leq 1.72 \pm 0.03$-fold, $p < 0.0001$) (Supplementary Fig. 15) and had higher change in responsiveness to variations in high-antigen density targets ($\leq 3.2 \pm 0.8$-fold, $p < 0.0001$), especially for NFAT (Supplementary Figs. 15, 19). Conversely, the 4-1BB CAR tended toward greater responsiveness to low-antigen targets, such as $P_{6xLexoOLEU2}$-CD19, especially for NF-κB (Supplementary Figs. 15, 19). AP-1 activity was more responsive to target variations in CD28 CAR T cells and was preferably induced by NALM6 ($p < 0.0001$) (Supplementary Figs. 16,19).

This type of CAR response characterization is normally done using antigen presentation on non-cellular materials[34,41,42,48–52]. Hence, we next compared CAR activation by SCASA yeast with CD19-coated microbeads. Baseline CD19 expression in yeast reached ≤3,606 CD19 molecules per cell with current strain designs without GPCR stimulation, while loading of microbeads saturated at ≤385,608 CD19 molecules per bead (Fig. 4c, Supplementary Fig. 20). As a clinical reference, a study reported patient DLBCL cells to have CD19 expression levels of 5810 CD19/cell, which dropped to 2,021 CD19/cell in relapsed cells following CAR T cell therapy (axicabtagene ciloleucel)[31]. Microbeads could provide higher maximum activation than yeast for NF-κB, NFAT, and AP-1 ($p < 0.0001$), with the biggest difference being $1.59 \pm 0.09$-

fold between yeast and microbeads for the 4-1BB CAR NF-κB response (Fig. 4c, Supplementary Fig. 21). However, yeast was more efficient than microbeads at activating CARs. This was evident from the significantly higher CAR activation per CD19 molecule of yeast (Fig. 4c, Supplementary Fig. 22), as exemplified by the $28.9 \pm 2.6$-fold higher activation per CD19 of $P_{PGK1}$-CD19 (917 CD19/cell) relative to 2 μg/mL CD19 microbeads (29,744 CD19/microbead) ($p < 0.0001$) averaged across all parameters (Supplementary Figs. 21, 22). Similarly, yeast-based activation using CD19 was $64.3 \pm 14$-fold more potent than microbeads, as evident from the dose-response $EC_{50}$-values (Fig. 4c). For example, $P_{TDH3}$-CD19 yeast (3,607 CD19/cell) induced similar CAR activation to 4 μg/mL CD19 microbeads (81,901 CD19/microbead) (Supplementary Fig. 21). Yeast and microbeads could equally determine the fold differences in NFAT, NF-κB, and AP-1 response dynamics for the 4-1BB and CD28 CARs in relation to antigen density variations (Supplementary Fig. 23). In addition to microbeads, we employed CD19-coated flat-bottom microtiter plates for CAR activation (Supplementary Figs. 21, 23), which also provided resolution for comparing 4-1BB and CD28 CAR responses (Supplementary Fig. 23), but with significantly higher activation than yeast and microbeads (Supplementary Fig. 21). However, the planar and continuous plate surface was physically distinct from the discrete amount of individual spherical yeast cells and microbeads, which also prevented control of both target-to-effector ratios and the amount of CD19 molecules per target.

In comparison to NALM6 cancer cells, both CAR designs could respond with equal or higher intensity (Fig. 4b), dynamic range (Supplementary Fig. 16, 17), and responsiveness (Supplementary Fig. 19) to CD19+ SCASA yeast. Yeast itself ($P_{PGK1}$-Empty) showed orthogonality to T-cell activation, as it did not activate NFAT, NF-κB, or AP-1 for either the CD28 or 4-1BB CAR (Fig. 4, Supplementary Figs. 15–19). Notably, yeast could cover a physiologically relevant range of CAR stimulation without increasing cell ratios, from the incipient responses to $P_{FUS1}$-CD19 to the activation by $P_{TDH3}$-CD19 comparable to NALM6, for both NF-κB and NFAT (Fig. 4b, Supplementary Figs. 17, 19). Notably, the $P_{MFA2}$-CD19 design alone could significantly resolve the main response dynamics (Fig. 4, Supplementary Figs. 15–19), demonstrating the feasibility of a single-strain approach for general CAR design characterization. Lastly, the CARs showed similar response patterns to PRP-dependent $P_{FUS1}$-CD19 and non-shmooing PRP-orthogonal $P_{6xLexoLEU2}$-CD19 (Fig. 4b, Supplementary Figs. 16, 17,19 and Supplementary Fig. 8), confirming that responsiveness is a matter of antigen-density increase and not PRP activation.

In summary, using SCASA yeast cells, we determined that two CAR designs affect NF-κB, NFAT, and AP-1 activity in three different ways, with dynamics varying with CAR-stimulation intensity, of which antigen density and target cell ratios are individual parameters. The CD28 CAR was generally more responsive and exhibited stronger responses than the 4-1BB CAR. Our conclusions regarding these CAR designs, derived using yeast, align with the current knowledge in the field[26,38,44,47,90]. Importantly, this demonstration highlights yeast cells as a potent and effective alternative to antigen-coated microbeads and planar surfaces, as well as cancer cell lines, for modeling antigen presentation and characterizing CAR T cell responses.

## Characterizing a donor-derived CAR T cell product using SCASA yeast cells

One important aspect of advancing cellular immunotherapies is the ability to robustly characterize responses of CAR T cell products resulting from novel designs or manufacturing methods. For this purpose, we proceeded to demonstrate the potential of applying SCASA yeast cells by characterizing an alternative CAR T cell product from a healthy donor made using a CRISPR-MAD7 method[91] and based on the Hu19-CD8α-CD28-CD3ζ CAR that contains fully-human antigen-binding domains and has been associated with lower toxicity[32,92]. To

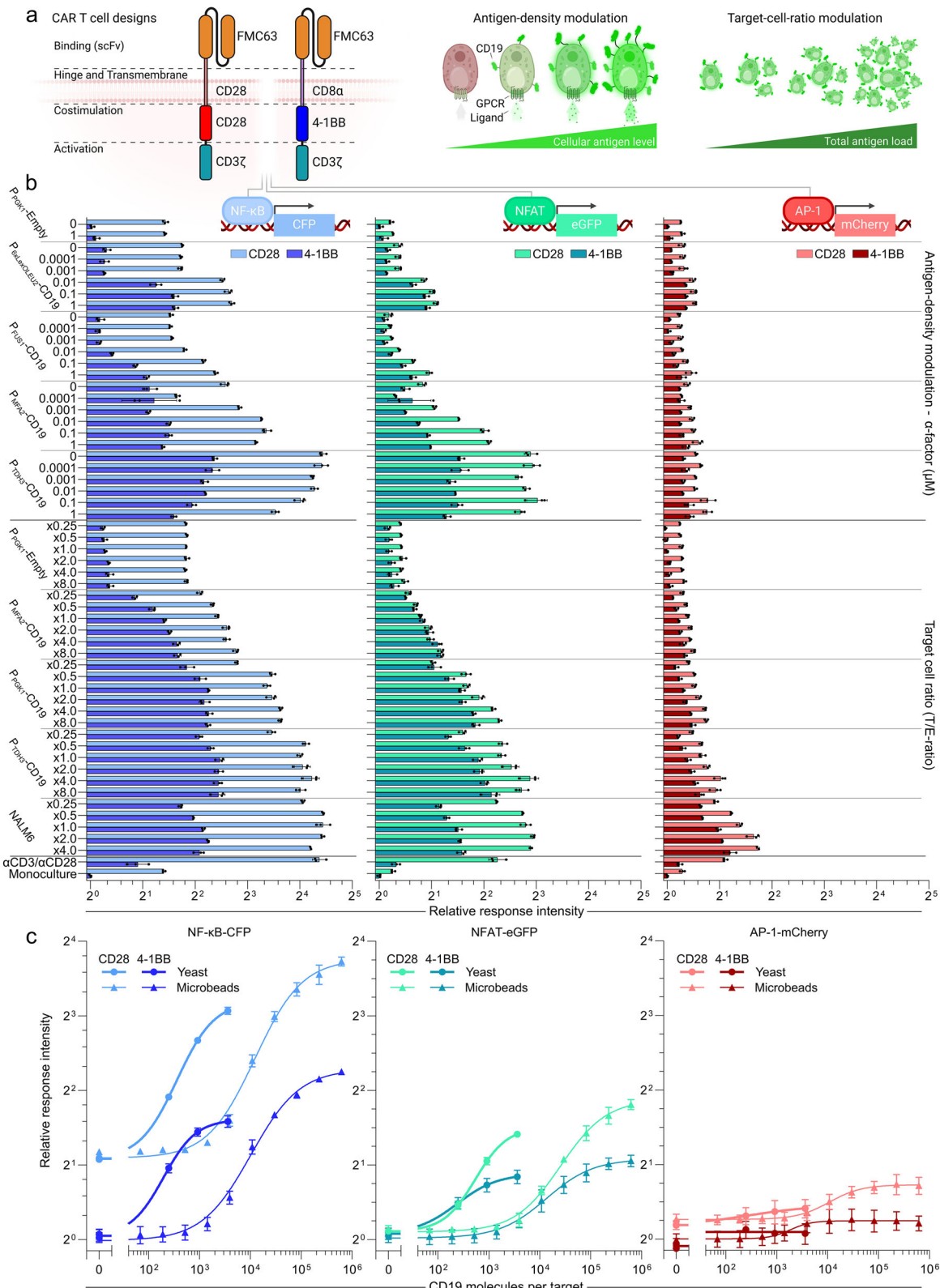

proceed testing with a single medium-range yeast design for cell product evaluation, $P_{PGK1}$-CD19 was selected as a suitable SCASA yeast example (Figs. 2g, 3c, 4b).

The $P_{PGK1}$-CD19 design activated the CAR T cells in a manner analogous to NALM6 across all examined target-to-effector cell ratios, hence verifying successful yeast-based simulation of CD19+ cancer cells towards the clinical Hu19-CD8α-CD28-CD3ζ CAR T cell product

(Fig. 5a–c). Specifically, CAR+ CD3+ T cells showed up-regulation of activation marker CD69 in co-cultures for all target cell ratios with $P_{PGK1}$-CD19 yeast ($p < 0.0001$) and NALM6 ($p \leq 0.0001$) (Fig. 5a-b). Meanwhile, no CAR- CD3+ T cells were activated for any target cell or condition, as also seen for the non-engineered control (CTRL) T cells (Fig. 5a, b). Furthermore, for the additional yeast designs $P_{MFA2}$-CD19 and $P_{TDH3}$-CD19, up-regulation of CD69 and CD25 was detected after

**Fig. 4 | Characterization of FDA-approved CAR T cell designs using SCASA yeast cells. a** Chimeric antigen receptor (CAR) T cell designs FMC63-CD28-CD28-CD3ζ (axicabtagene ciloleucel/brexucabtagene autoleucel) ('CD28 CAR') and FMC63-CD8α–4-1BB-CD3ζ (tisagenlecleucel) ('4–1BB CAR'), expressed in a triple-parameter-reporter (TPR) Jurkat T-cell line that couple T-cell activation transcription factors to fluorescent outputs; NF-κB-CFP, NFAT-eGFP, and AP-1-mCherry. CAR TPR Jurkats were co-cultivated for 24 h. with six CD19 SCASA yeast strains with different processing modules and hence different baseline CD19 levels. Antigen-density modulation was examined by G protein-coupled receptor (GPCR) stimulation of individual strains (α-factor) at a 1.0x target-to-effector (T/E) cell ratio. The effects of increasing target cell numbers were examined by modulating the T/E-ratio of SCASA yeast strains with fixed antigen densities and NALM6 from x0.25 to x8.0 relative to a fixed amount of CAR T cells. Created in BioRender. Deichmann, M. (2025): https://BioRender.com/65vslu2. **b** Relative response intensity of NF-κB-CFP (*blue*), NFAT-eGFP (*green*), and AP-1-mCherry (*red*) for the CD28 CAR (*light*) and

4-1BB CAR (*dark*) across all examined conditions, normalized to 4-1BB mono-cultures (log$_2$-scale). αCD3/αCD28 dynabeads (1.0x) were employed as a positive control for reporter genes. **c** Comparison between yeast and microbeads for CAR activation. CAR activation relative to the amount of CD19 molecules per target (log$_{10}$-scale) with comparison between equal numbers of SCASA yeast cells (*circle*) and CD19-coated microbeads (*triangle*) (1.5x T/E-ratio), measured by relative response intensity (log$_2$-scale). Microbeads had a 5.5 μm diameter, were streptavidin coated, and loaded with biotinylated CD19 (Supplementary Fig. 20). A comparison to CD19-coated microtiter plates was also done, however, the planar and continuous surface is physically distinct from yeast and microbeads, as well as with unknown antigen densities (Supplementary Fig. 21). Data represents means of median fluorescence intensities (mMFI) for three biological replicates ($n = 3$) and standard deviations hereof. All statistics and extended analyses in: Supplementary Figs. 15–24 and Supplementary Data 9–13. Source data are provided as a Source Data file.

---

5 months of cryopreservation of the CAR T cells in liquid nitrogen (Supplementary Fig. 25).

In summary, the CAR T cell product was functional towards CD19+ targets, as T cells activated solely upon CAR expression resulting from successful insertion via the CRISPR-MAD7 method. This was equally validated by the use of SCASA yeast cells or NALM6 cancer cells as targets (Fig. 5a–c).

## SCASA yeast cells are efficient and robust CAR T cell activators

Having confirmed that CAR T cell products could be verified with SCASA yeast, we next sought to investigate the activation efficiency, i.e. target cell capacity to induce responses, and robustness, i.e. target cell maintenance of stimulatory conditions. Here, we found that increasing the ratio of SCASA yeast cells per CAR T cell intensified CAR activation, with significantly increased CD69 expression levels per CAR T cell, rising from $13.3 \pm 0.5$-fold to $59.2 \pm 4.0$-fold over the level of CTRL T cells ($p < 0.0001$)(Fig. 5b), as well as enlargement of the population of activated CAR T cells (%CD69+) from $63.6 \pm 0.5\%$ to $84.7 \pm 0.6\%$ ($p < 0.0001$) (Fig. 5c). A similar behavior was seen for the response to NALM6 at increasing ratios, however, with significant differences to SCASA yeast, as NALM6 provided lower or equal intensity of CAR activation, spanning from $6.1 \pm 0.3$-fold to $44.6 \pm 1.4$-fold CD69 expression ($p < 0.0001$) (Fig. 5b), and had a larger spread in the number of activated CAR T cells, spanning from $50.3 \pm 0.9\%$ to $91.9 \pm 0.3\%$ ($p < 0.0001$)(Fig. 5c). Overall, SCASA yeast and NALM6 co-cultivations shared the conclusion that CAR T cells are activated in a cell-to-cell ratio-dependent manner in relation to the amount of activated CAR T cells and their response intensity.

Interestingly, P$_{PGK1}$-CD19 SCASA yeast cells could activate CAR T cells more intensely than NALM6 (Fig. 5b), yet had a CD19 level of merely $23.35 \pm 1.65\%$ compared to that of NALM6 (Supplementary Fig. 26). This higher efficiency of SCASA yeast to activate the CAR T cells led us to investigate the target cells' performance and robustness throughout co-cultivation. Here, we observed that NALM6 CD19 levels showed high instability throughout co-cultivation, evidenced by the occurrence of a CAR T cell-resistant CD19$^{low}$ relapse-like phenotype of the NALM6 cancer cells during co-cultivation, as also observed by others[26,93–95] (Fig. 5d). The appearance of the CD19$^{low}$ NALM6 phenotype was fully dependent on CAR expression (SV = 98.2%, $p < 0.0001$) (Supplementary Data 14), and hence developed only in CAR T cell co-cultivations, while being undetectable in NALM6 monocultures and in co-cultivations with CTRL T cells (Fig. 5d). Specifically, NALM6 CD19 levels dropped $98.54 \pm 0.05\%$ within 20 h. of co-cultivation to a level that was $55.4 \pm 2.2$-fold lower than in the CTRL co-cultivation ($p < 0.0001$)(Fig. 5e), resulting in a large alive CD19$^{low}$ population ($19.8 \pm 0.9\%$ of all cells detected) for the 5.0x co-cultivation with CAR T cells (Fig.5d, Supplementary Fig. 27). When equal amounts of NALM6 and CAR T cells were co-cultivated the effect was less drastic ($29.0 \pm 1.9$-fold lower CD19 levels) (Fig. 5e), resulting in killing of most NALM6 cells ($0.5 \pm 0.1\%$ of all cells detected) (Supplementary Fig. 27). When CAR T cells were in

excess, all NALM6 cells were killed (Supplementary Fig. 27). Comparatively, SCASA yeast cells showed up to 26.5-fold more robust CD19 presentation than NALM6 throughout co-cultivations (Fig. 5d, e), with only $2.1 \pm 0.1$-fold to $2.8 \pm 0.1$-fold differences in CD19 levels between CTRL and CAR T cell co-cultivations (Fig. 5e). Additionally, CD19+ SCASA yeast cell numbers were up to 17.0-fold more robustly maintained during CAR T cell co-cultivations than observed for NALM6 cells ($p < 0.0001$) (Fig. 5f, Supplementary Fig. 28), and there was no indication that SCASA yeast cells were killed by the CAR T cell cytotoxic response, contrary to NALM6 (Supplementary Fig. 27). Hence, the difference in CAR T cell activation patterns could be explained by a higher robustness in both antigen density and cell numbers of SCASA yeast cells compared to NALM6 (Fig. 5d–f), allowing yeast to be a more consistent target cell population to continuously provide activation signals to the CAR T cells, resulting in a higher efficiency of activation (Fig. 5a–c).

Further inspection of the CAR T cells showed that CAR expression patterns and the amount of CAR+ cells per CD3+ T cells alternated throughout co-cultivation in a CD19-dependent manner. CAR expression was elevated at low and equal target-to-effector cell ratios ($p < 0.0001$), while target cell outnumbering of CAR T cells was associated with lower CAR expression intensities ($p < 0.0001$) (Fig. 5g). This antigen-dependent up- and down-modulation of CAR surface levels has previously been shown to be dependent on CAR-stimulation levels, and reported to be caused by at least CAR expression regulation and CAR internalization[26,34,41,47,93,96]. The %CAR+ of CD3+ T cells dropped with an increase in target cell numbers ($p < 0.0001$), indicating a loss of CAR+ T cells (Fig. 5h). Importantly, these effects were seen for both SCASA yeast cells and NALM6, but were in both cases more pronounced for NALM6 (Fig. 5g, h).

Orthogonality was confirmed by the CD19$^{neg}$ yeast control (P$_{PGK1}$-Empty), which did not activate CAR T cells or CTRL T cells under any circumstances, showing no increase in CD69 or CD25 (Fig. 5a, b, Supplementary Fig. 25), did not affect CAR expression intensity (Fig. 5g), or change the number %CAR+ cells per CD3+ cells (Fig. 5h), collectively showing that the activation of CAR T cells by SCASA yeast cells was specifically caused by the interaction between surface-displayed CD19 and successfully expressed CARs, and that yeast could not activate CD3+ T cells unspecifically, corroborating the CAR Jurkat observations (Figs. 3, 4).

In summary, SCASA yeast cells provided an efficient method for activating and confirming functionality of a donor-derived CAR T cell product. SCASA yeast cells performed on par with cancer cells in terms of CAR T cell activation, yet with higher robustness and consistency regarding antigen density and target cell numbers than cancer cells throughout co-cultivation (Fig. 5i).

## Discussion

Currently, CAR T cells are one of the most expensive types of therapy, leading to inequality in accessibility and rendering it

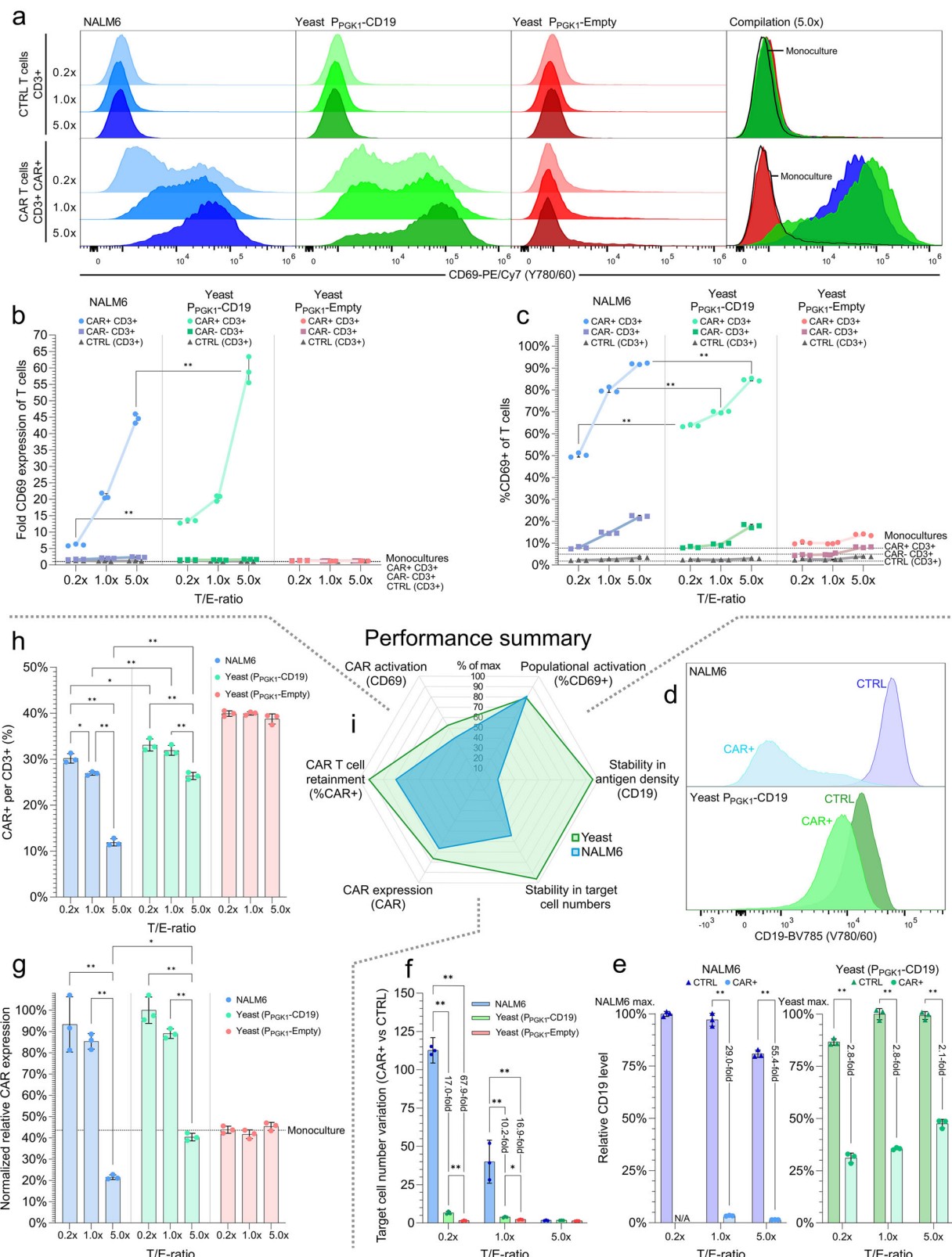

prohibitively expensive for healthcare systems[97]. The CAR field is still emerging, and major challenges need to be mitigated to improve safety and efficacy, including mitigation of inhibitory immunosuppression, severe toxicity, CAR T cell trafficking, and target antigen escape[3,30,98]. Hence, to manifest the full potential of CAR T cells, it is essential to improve CAR product performance and develop safer and more cost-effective cellular designs, supported by

innovative ways to enhance our understanding of the obstructive mechanisms.

Various experimental strategies have been employed to understand how antigen density and modulation impact CAR T cell responses, each with distinct advantages and limitations. Typically, these employ target cell panels with differential antigen densities enabled by either exploiting default antigen variation in cell lines[40,45],

**Fig. 5 | Activation of CAR T cells from a donor using SCASA yeast cells. a** CD69 expression in non-engineered control (CTRL) T cells (alive, CD3+) (*upper*) and chimeric antigen receptor (CAR) T cells (alive, CD3+, CAR+) (*lower*) after co-cultivation with NALM6 (*blue*), SCASA yeast $P_{PGK1}$-CD19 (*green*), and negative control yeast $P_{PGK1}$-Empty lacking CD19 (*red*), at different target-to-effector (T/E) cell ratios (0.2x, 1.0x, 5.0x) for 20 h. An overlay for 5.0x co-cultivations is shown (*far right*), including unstimulated T cell monocultures (*black line*). **b** CD69 expression intensity of co-cultivation populations: CAR+ (*circle*) and CAR- (*square*) T cells from the CAR-engineered population, CTRL T cells (*triangle*), and baseline CD69 of monocultures (*dotted lines*). Normalization: CTRL T cell monoculture. **c** Percentage of T cell populations expressing CD69 (%CD69+) at any intensity for co-cultivations, and baseline %CD69+ of monocultures (*dotted lines*). **d** Target cell CD19 levels after co-cultivations at 5.0x for NALM6 and $P_{PGK1}$-CD19 yeast with CAR T cells (*light*) and CTRL T cells (*dark*). **e** CD19 levels relative to maximum of NALM6 and $P_{PGK1}$-CD19 yeast individually after co-cultivations with CTRL T cells (*triangle*) and CAR

T cells (*circle*). **f** Fold target cell number variation per alive CD3+ T cell between CTRL and CAR T cell cultures for NALM6, $P_{PGK1}$-CD19, and $P_{PGK1}$-Empty, disregarding increased or diminished levels of cells (reciprocal value of fold changes<1). **g** CAR expression intensity of CAR+ population relative to CAR- population (background), normalized to maximum. **h** Percentage of CAR+ T cells per CD3+ T cell detected after co-cultivations. **i** Comparison of general performance of $P_{PGK1}$-CD19 SCASA yeast and NALM6 cancer cells, quantified as percentage of most extreme behavior observed (% of max), averaging across T/E ratios. Summary of all other plots (**a**–**h**). Data represents means of cell counts or median fluorescence intensities (mMFI) for three biological replicates ($n = 3$) and standard deviations hereof. Histograms are representative replicates and normalized to mode (CD69) or unit area (CD19). Selected comparisons from two-way ANOVAs with Tukey's multiple comparisons tests are shown. Significance levels: *$p \leq 0.05$, **$p \leq 0.001$. Statistics and extended analyses in: Supplementary Fig. 25–28 and Supplementary Data 14. Source data provided as a Source Data file.

engineering antigen-gene copy-numbers[26,33–36], differential target-antigen mRNA electroporation[39,46], or modular cell platforms for transiently attaching purified antigens[47]. Mammalian cell line approaches have the great advantage of close imitation of in vivo cellular interactions, but can require laborious multiplex engineering and have risks of genetic drift[47,53]. Importantly, mammalian cell lines are not orthogonal and adapt to CAR T cell exposure. This is exemplified by their cytotoxic sensitivity resulting in target cell loss biased by antigen density or defensive responses to this selective pressure, such as rapid and extensive antigen modulation[18,26,93–95]. This leads to inconsistency in primary parameters of CAR-responses, antigen density and target cell numbers, even down to the hourly timescale in both in vitro and in vivo experimental models[26,93–95], as also seen in this study for NALM6 (Fig. 5d–f, Supplementary Figs. 27, 28). We have shown that SCASA yeast cells are more resistant to these distorting effects and remain robust signal providers upon CAR T cell exposure (Figs. 3, 4, 5, Supplementary Figs. 13, 14, 27, 28).

Similarly, to address mammalian cell line challenges, robust non-cellular material-based systems, such as microbead-bound antigen, have been employed to ensure precise antigen control and to orthogonalize the activating molecular interactions from other cellular interactions and confounding effects[34,41,48–53]. We have shown that overall CAR T cell activation dynamics (NF-κB, NFAT, and AP-1) and distinct CAR design behaviors (CD28 and 4-1BB CARs) can be comparably recapitulated using both yeast cells and non-cellular antigen platforms, such as microbeads and microtiter plates (Fig. 4, Supplementary Fig. 21). However, compared to commercial antigen platforms, yeast can offer a more cost-effective alternative, as self-cultured yeast can serve as low-cost reagents for CAR T cell activation with high effectiveness (Figs. 4, 5). In addition, the fully DNA-encoded yeast platform is independent of both handling costly purified proteins and the reliability of their surface attachment. Importantly, yeast is not limited by material quantity, as large amounts of antigen-expressing yeast can be rapidly grown, which can reduce both experimental risk and cost, can remove time pressure associated with the limited stability of purified proteins, can enable large-scale experiments (scalability), and allows room for experimental optimization.

Orthogonality is arguably also a limitation[53], as non-cellular systems and SCASA yeast likely do not simulate the full spectrum of cellular interactions between cancer and immune cells. Hence, mammalian cell lines are arguably better suited for more holistic studies of the complex interplay between CAR T cells and cancer cells, such as in cytotoxicity assays[26,38,53]. CAR-based T-cell activation depends on the physical dimensions of the intercellular space and supramolecular immunological synapse structures, favoring kinase activity and excluding phosphatases[52,99,100]. However, CAR-synapses are more disorganized than conventional T-cell synapses, differ in employment of co-stimulatory, adhesion, and accessory receptor molecules, and are additionally affected by CAR affinity and target cell

antigen density[41,42,52,99,100]. SCASA yeast relies on the ability of antigen-density model systems to be greatly simplified, analogously to non-cellular platforms, as CAR T cells do not exploit accessory receptors as conventional T cells[41,47,52]. For example, while LFA-1-binding ICAM-1 and CD2-binding CD58 have been shown to greatly increase TCR sensitivity against non-cellular systems, the effect is only modest on CAR T cells[41,47]. Accordingly, yeast-based systems displaying pMHC-II for TCR activation were initially less successful[64], but has since been functionalized by co-display of ICAM-1[60]. This indicates that while oversimplification is a risk of orthogonality, yeast inherently provides a solution by being a flexible platform for bottom-up engineering of complexity to approximate native interactions, while retaining control[59,60].

Lastly, absolute orthogonality could potentially be difficult to obtain when employing yeast, as T cells express pattern recognition receptors (PRR) that can bind pathogen-associated molecular patterns (PAMP), such as Toll-like receptor 2 (TLR2) and CD5 recognition of yeast cell wall-derived zymosan, which upon stimulation can modify T cell responses[101–104]. Interestingly, yeast-induced PRR stimulation of T cells might induce relevant phenotypes to cancer immunotherapy, with for example studies showing that TLR2 stimulation can improve T cell anti-cancer phenotypes[105], lower antigen density thresholds for T cell activation[106], and improve CAR T cell responses[107]. The possible impact on synapse formation, lack of accessory molecules, physical dimensions, and potential T cell PRR-engagement by yeast PAMPs on CAR T cell activation are considerable uncertainties to be addressed in future investigations. Nevertheless, we did not encounter any hindrances of using yeast during co-cultivation or activation of T cells.

We have demonstrated that SCASA yeast can effectively simulate CD19+ cancer cells with precise on-demand antigen densities that control the activational state of CAR T cells, allowing for high-resolution interrogation of CAR T cell responses (Figs. 3–5). Specifically, we employed yeast to characterize CAR sensitivity to changes in antigen density and cellular ratios (Figs. 3–5), to verify that the response of the CD28 CAR is more intense than the 4-1BB CAR, particularly with high-antigen density targets, and that these CAR designs differentially affect NFAT, NF-κB, and AP-1 activities, which indicates patient-specific performance (Fig. 4), and, finally, to validate the functionality of a CAR T cell product made from donor blood using an alternative CRISPR-MAD7 method and CAR design (Fig. 5). In addition to CAR design characterization and cell product evaluation, applications could include the expansion of CAR-specific T cell populations during manufacturing[108].

The SCASA yeast platform has potential for further advancement. We anticipate that optimizing antigen expression could increase surface densities and, in turn, enhance CAR activation to the peak levels observed with antigen-coated microbeads and microtiter plates. Further PRP engineering may enable tighter and more dynamic control of antigen density. Exchangeable GPCRs can be exploited to link

responses to endogenous ligands (Fig. 2), to ensure ligand orthogonality to target cells, for biomarker sensing, or to build cell-cell feedback-regulated systems[1,56,71]. The system is compatible with multiplexing through additional effector cassettes, and could allow incorporation of additional layers of orthogonal signaling inputs[79]. Similarly, co-display allows for investigation of multiple signals and closer imitation of immuno-oncological cellular interactions[41,47,60]. Furthermore, the fully DNA-encoded SCASA system results in a genotype-phenotype bond compatible with evolution platforms and screening of variant libraries[56,65,66].

In summary, we consider yeast to have great potential as an effective, robust, and customizable platform for investigating the dynamics of specific molecular cell-cell interactions with the ability to combinatorially modify and control signal molecules in a high-throughput manner. With the demonstrated potential and engineerability of yeast platforms, as well as developments and trends in yeast synthetic biology, we anticipate the advancement of yeast-based systems toward targeting immunological challenges and providing novel applications in the future.

## Methods

### Molecular cloning for yeast engineering

**PCR and DNA handling.** Genomic DNA (gDNA) was purified using the Yeast DNA Extraction Kit (Thermo Scientific, Cat.#78870). Custom DNA synthesis was done using gBlocks™ Gene Fragment synthesis service (Integrated DNA Technologies). For amplification of DNA fragments for cloning or genome integration, and inverse PCR plasmid construction, Phusion High-Fidelity (HF) PCR Master Mix with HF Buffer (Thermo Scientific, Cat.#F531L) was used, and, specifically, for amplification of Uracil-Specific Excision Reagent (USER) cloning DNA fragments, Phusion U Hot Start PCR Master Mix (Thermo Scientific, Cat.#F533L) was used with primers with USER-compatible uracil-containing tails. Standard 50 μL PCR reactions were composed of; 25 μL PCR Master Mix (2X), 2 μL of each primer (10 μM), DNA template scaled to 100–200 ng gDNA or 10–20 ng plasmid DNA, and Milli-Q H$_2$O to a total reaction volume of 50 μL. For diagnostic and genotyping PCRs, such as colony PCR of *Escherichia coli* or yeast, One*Taq*® Quick-Load® 2X Master Mix with Standard Buffer (New England BioLabs, Cat.#M0486) was used. For each *E. coli* colony PCR, a 10 μL PCR mix was made; a small amount of cell mass was added to a mix of 5 μL PCR Master Mix (2X), 1 μL of each primer (10 μM), and 3 μL Milli-Q H$_2$O. For each yeast colony PCR, a 20 μL PCR mix was made; cell mass was added until visible turbidity in a mix of 1 μL of each primer (10 μM) and 8 μL Milli-Q H$_2$O, followed by cell lysis by heating at 98 °C for 15 min., and after cooling, 10 μL PCR Master Mix (2X) was added. PCRs were conducted according to the manufacturer's protocols using an S1000 Thermal Cycler (Bio-Rad). All primers were synthesized using a Custom DNA Oligos service (Integrated DNA Technologies) and can be found in Supplementary Data 15. Gel electrophoresis was done in 1%w/v agarose gels of 1X Tris-acetate-EDTA buffer at 90 v for 30 min., using TriTrack DNA 6X Loading Dye (Thermo Scientific, Cat.#R1161), GeneRuler 1 kb DNA Ladder (Thermo Scientific, Cat.#SM0311), and RedSafe™ Nucleic Acid Staining Solution (iNtRON Biotechnology, Cat.#21141). Gel imaging was conducted on a Gel Doc XR+ System (Bio-Rad). PCR amplicons were purified via gel purification or column purification using the NucleoSpin Gel and PCR Clean-up kit (Macherey-Nagel, Cat.#740609.50). Amplicon sequences were verified by Sanger sequencing using the Overnight Mix2Seq Kit service (Eurofins Genomics). Primers, DNA fragments, amplicons, plasmids, and gDNA were always dissolved in Milli-Q H$_2$O and stored at -20 °C. An overview of all DNA parts, backbones, and plasmids can be found in Supplementary Data 16-17.

**Construction of plasmids.** Plasmid vectors containing genome-integration expression cassettes were based on the EasyClone-

MarkerFree system compatible with CRISPR/Cas9 engineering[109], and were constructed with USER cloning and assembly techniques[110], however, modified by using custom standardized USER-compatible primer tails for DNA parts and inverse PCR for conversion of USER-compatible backbones, denoted MAD-cloning (Supplementary Fig. 29). Linearized USER-ready MAD-cloning backbones were generated by purification of inverse PCR-amplified EasyClone-MarkerFree integrative vectors using primers with USER-compatible tails. Similarly, custom yeast expression plasmids were constructed by converting template expression plasmids[111,112] into MAD-cloning compatible backbones. Employed parts, such as promoters, terminators, protein-coding sequences (CDS), or entire genes, were amplified from yeast gDNA, plasmids, or ordered as gBlocks™, and in some cases modified by in-frame fusion using USER-cloning, and coding sequences were modified to contain the AAAACA Kozak sequence. Each cassette assembled by MAD-cloning contained standardized interchangeable parts: I) a promoter, II) a CDS, III) a terminator, IV) a linearized backbone, which for genome integration contained homology arms directed at EasyClone-MarkerFree genome integration sites flanked by *Not*I-sites. USER-assembly reactions contained 20–40 ng linearized USER-ready backbone, 40-80 ng of each USER-compatible part, 0.5 μL USER® Enzyme (New England BioLabs, Cat.#M5505), 1 μL 5X Phusion HF Buffer (Thermo Scientific, Cat.#F518L), and Milli-Q H$_2$O to a total reaction volume of 5 μL. USER-assembly mixes were incubated at 37 °C for 30 min. and transformed into *E. coli*. Plasmids for integration of GPCR expression cassettes were assembled as previously described[67]. For CRISPR/Cas9 engineering, all gRNA was expressed from cassettes using the snoRNA *SNR52* promoter and a *SUP4* terminator[113]. Custom yeast gRNA-expression plasmids were generated by inverse PCR of template gRNA-expression plasmids using a 5′-phosphorylated primer and a primer with a tail for replacing the 20 bp gRNA target sequence. Purified amplicons were then blunt-end ligated in a reaction containing: 1 μL T4 DNA Ligase (Thermo Scientific, Cat.#EL0011), 2 μL 10X T4 DNA Ligase Buffer (Thermo Scientific, Cat.#EL0011), and 17 μL amplicon dissolved in Milli-Q H$_2$O, which was incubated for 2 h. at room temperature. Template plasmid was then removed by addition of 1 μL DpnI (New England BioLabs, Cat.#R0176), 5 μL 10X FastDigest Buffer (Thermo Scientific, Cat.#B64), and 24 μL Milli-Q H$_2$O, which all was incubated for 1 h. at 37 °C, and finally transformed into *E. coli*. All *E. coli* transformants for plasmid assembly were genotyped by colony PCR to screen for successful assembly, and correct assembly was verified by Sanger sequencing of purified plasmids using the Overnight Mix2Seq Kit service (Eurofins Genomics). Plasmids were purified from *E. coli* cultures using the NucleoSpin Plasmid kit (Macherey-Nagel, Cat.#740588.50). All plasmids contained ampicillin resistance (*ampR*) for propagation in *E. coli*, and retention in yeast relied on auxotrophic markers (*HIS3*, *URA3*, *LEU2*, *TRP1*) or nourseothricin antibiotic resistance (*NatR*). Primers, DNA fragments, amplicons, plasmids, and gDNA were always dissolved in Milli-Q H$_2$O and stored at −20 °C. An overview of parts, backbones, constructed plasmids, and plasmids originating from other studies[67–69,72,73,83,84,109,111,112] can be found in Supplementary Data 16-17, and primers in Supplementary Data 15.

**Repair and Intentional Primer Dimer (IPD) PCR templates for gene knockout or alteration.** For direct genome engineering, such as gene knockouts or minor alterations of genome sequences, DNA repair templates were co-transformed with gRNA-plasmids into Cas9-expressing yeast strains. Repair templates were either two genome-amplified sequences with overlapping homology defined by primer tails or an Intentional Primer Dimer (IPD) PCR amplicon with flanking homology to the cut site. IPD-PCR templates were generated by annealing and PCR-amplifying two complementary primers using each other as templates at their predicted annealing temperatures (50 μL Phusion HF, with 4 μL of each primer at 10 μM). IPD-PCR templates had genome-targeting homology tails and contained a complementary

binding region for their amplification, which was also composed of one of several genome-unique CRISPR landing pads previously characterized[114], allowing for easy reengineering of the site (Supplementary Fig. 30).

**Bacterial handling for plasmid construction and propagation.** Plasmid propagation and USER-assembly were conducted by heat-shock transformation into *E. coli* strain DH5α, which was cultivated in Terrific Broth or Luria-Bertani with ampicillin (100 mg/L) at 37 °C in liquid media at 300 r.p.m. or on agar plates for 16–20 h. Before transformation, *E. coli* DH5α was made chemically competent and stored at -80 °C until needed[115]. All *E. coli* strains were stored as cryostocks in media with 25% (v/v) glycerol at -80 °C.

## Yeast strains, media, and engineering
**Yeast handling.** All yeast strains were engineered from *S. cerevisiae* strain BY4741 (Supplementary Data 18). Strains were grown in YPD media with 2% (w/v) glucose except for during experiments or for plasmid retention, in which strains were grown in Synthetic Complete (SC) media with 2% (w/v) glucose using appropriate amino acid dropout mixes (Sigma-Aldrich) and/or with the addition of antibiotic nourseothricin (100 mg/L) (Jena Bioscience, Cat.#AB-101). For experiments requiring pH-control, a SC medium with ammonium sulfate (AS) and urea (SC-AS/Urea) was buffered using citrate-phosphate buffer (CPB)[116]. Yeast strains were generally grown at 30 °C at 250 r.p.m. or on 2% (w/v) agar plates, unless otherwise stated for specific experiments. All strains were stored as cryostocks in media with 25% (v/v) glycerol at -80 °C.

**Yeast transformation and engineering.** All transformations of yeast were done using the LiAc/ssDNA/PEG method[117]. Plasmid-based expression utilized either 2μ- or CEN/ARS-type plasmids. All genome engineering relied on CRISPR/Cas9-based methods, by having cells constitutively expressing Cas9 from pEDJ391[111]. Genome integrations relied on co-transformation of I) FastDigest™ *Not*I-linearized (Thermo Fisher, Cat.#FD0593) cassette-containing USER-assembled plasmids with homology arms and II) gRNA helper vectors, both targeting the characterized integration sites described for the EasyClone Marker-Free system[109]. Gene knockouts relied on co-transformation of I) a gRNA expression vector targeting the site of interest and II) an IPD-PCR repair template with homology flanking the Cas9 cut site, which contained a genome-unique CRISPR landing pad for enabling reengineering of the site[114]. Similarly, scarless gene alterations were done by using an IPD-PCR template containing the novel sequence of interest. Genomic engineering was confirmed by genotyping via colony PCR and Sanger sequencing of cassettes, as described above.

From the BY4741 background, the strain DIX14 was optimized for GPCR biosensing by overexpression of native α-factor pheromone-sensing mating GPCR Ste2 ($P_{CCW12}$-STE2-$T_{CYC1}$), and by balanced expression of native G protein $G_α$-subunit ($P_{PGK1}$-GPA1-$T_{CYC1}$), as well as by deletion of native PRP-coupled GPCRs (ste2Δ0 and ste3Δ0), native $G_α$-subunit (gpa1Δ0), a negative feedback regulator (sst2Δ0), an α-factor protease (bar1Δ0), and a cyclin-dependent kinase inhibitor that induce G1 cell cycle arrest (far1Δ0). Furthermore, native *AGA2* was deleted in preparation for YSD optimization (aga2Δ0). From DIX14, DIX17-28 were created by integration of yEGFP-expression cassettes. DIX34 was created by knocking out the PRP transcription factor Ste12 (ste12Δ0) and integrating a LexA-based synthetic transcription factor, as well as a yEGFP cassette with a LexO-containing promoter for orthogonalized signaling. The strain DIX41 was optimized for YSD by Aga1 overexpression ($P_{TDH3}$-AGA1-$T_{ADH1}$). DIX44-56,74 strains were generated from DIX41 for CD19 YSD experiments and co-cultivations, by integration of CD19-expression cassettes. Based on DIX45, genome alterations were made to replace the Gpa1 $G_α$-subunit with $G_{αz}$- and $G_{αi2}$-chimeras, to generate DIX57 and DIX58, respectively, for allowing

coupling of heterologous GPCRs. From these strains, heterologous GPCRs were integrated to make the GPCR strain library (DIX59-DIX65). The background strain for orthogonal GPCR signaling (DIX33) was equipped with Aga1 overexpression and a CD19 cassette with the LexO-containing promoter for orthogonal GPCR-based control of CD19-display (DIX67). An overview of strains can be found in Supplementary Data 18.

## Human cell engineering and cultivation
**Construction of anti-CD19 CAR Jurkat NFAT-Luc and CAR TPR Jurkat cell lines.** To investigate the activation of CAR T cells, a Jurkat cell line equipped with a nuclear factor of activated T cell (NFAT) transcription factor luciferase reporter system (Jurkat NFAT-Luc) (Nordic BioSite, Cat.#BPS-60621) was further modified by insertion of an anti-CD19 CAR (FMC63-CD8α-4-1BB-CD3ζ)[88], hence providing a bioluminescent luciferin signal of an intensity proportional to the level of T cell activation. Likewise, a triple-parameter-reporter Jurkat T cell line (TPR Jurkat cells) was employed to characterize CAR-induced activation of downstream transcription factor activity by fluorescent reporter genes, namely; NF-κB-CFP, NFAT-eGFP, and AP-1-mCherry[89]. Two different CAR designs were inserted into the TPR Jurkat cell line; FMC63-CD8α-4-1BB-CD3ζ (referred to as the '4–1BB CAR') and FMC63-CD28-CD28-CD3ζ (referred to as the 'CD28 CAR'). Lentiviral particles containing the CD19 CAR elements were produced by lipofectamine-based co-transfection of HEK293 cells (ATCC, 293 T Cat.#CRL-3216) with 3rd generation packaging plasmids pMD2.G (Addgene, #12259), pMDLg/pRRE (Addgene, #12251), pRSV-Rev (Addgene, #12253), and the CAR transfer vector (pDTU), a modified version of pLenti-puro (Addgene, #39481), optimized to include a cPPT-CTS sequence, the EF-1α promoter, and a Woodchuck Hepatitis Virus (WHV) Post-transcriptional Regulatory Element (WPRE). 24 h. after transfection, lentiviral particles were harvested and concentrated using Lenti-X concentrator (Takara Bio) and stored at -80 °C. Jurkat cell lines were transduced with lentivirus at an MOI of 5. After transduction, anti-CD19 CAR+ Jurkat NFAT-Luc and TPR Jurkat cells were single-cell sorted and expanded to adequate numbers before cryopreservation in liquid nitrogen. Jurkat cell lines were cultured in RPMI 1640 medium (ATCC modification) (Thermo Fisher, Cat.#A1049101) with 10% fetal bovine serum and 1% pen-strep (RPMI + 10%FBS + 1%PS) at 37 °C with 5% $CO_2$.

**NALM6 cancer cell line.** The human CD19+ NALM6 B cell precursor leukemia cell line (DSMZ, no.: ACC 128) was included as a benchmark positive control. NALM6 was cultured in RPMI + 10%FBS + 1%PS at 37 °C with 5% $CO_2$. The cell line was further engineered by genomic insertion of a CAG-promoter driven GFP-cassette in safe-harbor site AAVS1_3 using CRISPR-MAD7, as previously described in refs. 91,118.

**Isolation of T cells from peripheral blood and CRISPR-MAD7 engineering for CAR insertion.** This study was carried out in accordance with the Declaration of Helsinki. Human peripheral blood was obtained from healthy adults after obtaining informed consent (Technical University of Denmark - Rigshospitalet National Hospital approval BC-40). No personal information was gathered, and donors were anonymized for this study. All T cells were derived from the blood of healthy donors collected at the central blood bank at Rigshospitalet (Copenhagen, Denmark). Peripheral blood mononuclear cells (PBMC) were isolated from buffy coat, by blood filtration using a Falcon 70 μM Cell Strainer (Corning, Cat.#352350), followed by 2X dilution in Dulbecco's Phosphate Buffered Saline with 2% Fetal Bovine Serum (Stemcell Technologies, Cat.#07905), and finally, density centrifugation using SepMate™-50 (IVD) tubes (Stemcell Technologies, Cat.#85460) and Lymphoprep™ (Stemcell Technologies, Cat.#07811), according to the manufacturer's protocols. Pan T cells were isolated from PBMCs by negative selection, using the EasySep™ Human T Cell Isolation Kit (Stemcell Technologies, Cat.#17951), with EasySep™ Buffer (Cat.#20144) and EasySep™ Magnet,

The "big Easy" magnet (Stemcell Technologies, Cat.#18001), according to manufacturer's protocols. T cells were then resuspended at $1 \times 10^6$ cells/mL in ImmunoCult™-XF T Cell Expansion Medium (Stemcell Technologies, Cat.#10981) supplemented with Human Recombinant IL-2 (12.5 ng/mL) (Stemcell Technologies, Cat.#78036), IL-7 (5 ng/mL) (Stemcell Technologies, Cat.#78053), IL-15 (5 ng/mL) (Stemcell Technologies, Cat.#78031), and ImmunoCult™ Human CD3/CD28/CD2 T Cell Activator (25 µL/mL) (Stemcell Technologies, Cat.#10990) and cultured for 1.5 days at 37 °C with 5% CO$_2$. After activation, T cells were engineered using CRISPR-MAD7 according to a recently published method[91], to insert the CAR Hu19-CD8α-CD28-CD3ζ (Hu19-CD828Z), containing an anti-CD19 fully-human scFv (Hu19), CD8α hinge and transmembrane domains, a CD28 costimulatory domain, and a CD3ζ activation domain[92] (Clinical trial: NCT02659943). In addition, the CAR contained a myc-tag for the detection of surface expression. The CAR was expressed using the EF-1α promoter and a bovine growth hormone polyadenylation (bgh-PolyA) signal from the AAVS1_3 safe-harbor site. The CRISPR-MAD7 AAVS1_3 crRNA ribonucleoprotein formulation, Hu19-CD8α-CD28-CD3ζ homology-directed repair (HDR) template generation, and transfection of donor-derived (primary) T cells using a Lonza 4D-Nucleofector with shuttle unit with use of M3814 (Selleckchem) recovery followed the previously described methods[91,118]. Along with the CAR T cells, a control (CTRL) T cell product was established from the same T cell isolate, undergoing the same transfection protocol as the CAR T cells, however, without using a CAR HDR template. The CAR T cell product was cultivated in RPMI + 10%FBS + 1%PS at 37 °C with 5% CO$_2$ for experiments and cryopreserved as $10 \times 10^6$ cells/mL in 1 mL 90% heat-inactivated Human AB Serum (Sigma-Aldrich, Cat.#H4522) + 10% DMSO in liquid nitrogen. T cells for investigation of viability during co-culture with yeast were obtained from PBMCs isolated from healthy donor buffy coats, by density centrifugation using LymphoPrep™ Solution (Axis-Shield PoC, Cat.#1858) and cryopreserved at −150 °C in FBS + 10% DMSO. T cells were isolated from PBMCs by negative selection, using the EasySep™ Human T Cell Isolation Kit (Stemcell Technologies, Cat.#17951).

## Experimental procedures

**Flow cytometry.** Flow cytometric measurements were carried out using either a NovoCyte Quanteon 4025 Flow Cytometer System with a NovoSampler Q System (Agilent) or a CytoFLEX-S (V-B-Y-R) instrument (Beckman Coulter), according to the manufacturers' protocols. Specifications are provided in the description of individual experiments below. Gating and compensation were done using FlowJo™ v10.8.1 Software (BD Life Sciences). Gating strategies employed fluorescence minus one (FMO) controls, as well as the employment of negative controls, where possible, for determining true positive signals (e.g. yEGFP+ , CAR+ ) (Supplementary Fig. 32-33).

**Processing module promoter characterization.** For processing module promoter characterization, a transcriptome analysis was conducted on a previously acquired dataset[67] (Supplementary Fig. 4, Supplementary Data 1), from which candidate promoters were chosen. Each promoter was defined as the 1000 bp sequence upstream of the gene CDS, and USER-assembled with yEGFP and T$_{CPS1}$, to form reporter cassettes. Dose-response and phenotype features of strains containing the selected promoters, P$_{MFA2}$, P$_{FUS1}$, P$_{AGA2}$, P$_{FIG1}$, P$_{MFA1}$, P$_{YCL076W}$, P$_{HPF1}$, P$_{IME4}$, P$_{CSS1}$, P$_{TDH3}$, P$_{PGK1}$ and P$_{6xLexOLEU2}$, were then investigated upon GPCR stimulation via yEGFP expression (DIX17-28 and DIX34) (Fig. 2, Supplementary Fig. 5–9). The yeast strains were inoculated and pre-cultivated in 0.5 mL SC (24 h., 30 °C, 250 r.p.m.), then diluted 10-fold and further cultivated in SC (24 h., 30 °C, 250 r.p.m.), whereafter, for each replicate, cultures were diluted 50-fold by transfer to 200 µL volumes comprising a 10-fold dilution series of α-factor (0–100 µM) (peptide-seq.: WHWLQLKPGQPMY) (GenScript) in SC + 1%DMSO, and then stimulated for 5 h. (30 °C, 250 r.p.m.) (Supplementary Fig. 31). Flow cytometric measurement was done using a NovoCyte Quanteon

flow cytometer (Agilent), with cells suspended in 1X phosphate-buffered saline (PBS), and collection of 20,000 cells per replicate to quantify yEGFP levels (B525/45-A) and PRP-regulated morphological changes (shmooing) (SSC-A). The gating strategy is described in Supplementary Fig. 32. To approximate the effects of PRP activation on expression not directly related to transcriptional upregulation, SSC-normalization was performed by dividing individual fluorescence intensities by the corresponding SSC light scatter intensity. Each condition was examined in three biological replicates. A two-way ANOVA with Tukey's or Dunnett's multiple comparisons test was conducted for individual conditions, and to fulfill the assumptions of parametric tests, some data was log$_{10}$-transformed, all specified in Supplementary Data 2-3. Linear and non-linear variable slope (four parameter) regressions can be found in Supplementary Data 3-4.

**CD19 display characterization.** To investigate the different variants of CD19 modules (Fig. 2, Supplementary Figs. 3,10-12), yeast cells were grown and stimulated to present CD19 differentially, whereafter they were stained with antibodies and assessed by flow cytometry. For the investigation of the strain library with constitutive and dynamically regulated expression of CD19 (No YSD, P$_{PGK1}$-Empty, P$_{MFA2}$-CD19, P$_{FUS1}$-CD19, P$_{FIG1}$-CD19, P$_{MFA1}$-CD19, P$_{HPF1}$-CD19, P$_{TDH3}$-CD19, P$_{PGK1}$-CD19), DIX41-56 strains were inoculated and pre-cultivated in 0.5 mL SC (24 h., 30 °C, 250 r.p.m.), then diluted 10-fold and further cultivated in SC (24 h., 30 °C, 250 r.p.m.), whereafter, for each replicate, 50,000 cells were transferred to 300 µL volumes comprising a 10-fold dilution series of α-factor (0–100 µM) (GenScript) dissolved in SC + 1%DMSO, and stimulated for 20 h. (30 °C, 250 r.p.m.). For conventional YSD using strain EBY100 and pCT-plasmid-based expression[112] (Supplementary Fig. 1), strains were pre-cultivated in SC -trp, then transferred to 5 mL SC -trp +2%Gal for induction of YSD or +2%Gluc as a negative control (24 h., 30 °C, 250 r.p.m.). For the investigation of P$_{GAL1}$-titratability of plasmid-based (DIX41 + pMAD58) and genome-integrated CD19 YSD (DIX74) (Supplementary Fig. 2), strains were pre-cultivated in SC -ura and SC, respectively, then transferred to SC -ura and SC, respectively, with varying concentrations of galactose (0.0002%-2%Gal), using raffinose as compensating carbon source, and YSD was induced for 24 h. (30 °C, 250 r.p.m.). Strains were similarly grown in 2%Gluc as negative controls. For assessment of the GPCR library for inducing CD19 YSD using different types of ligands (Fig. 2, Supplementary Fig. 3), strains were pre-cultivated in 0.5 mL SC (24 h., 30 °C, 250 r.p.m.), then diluted 10-fold and further cultivated in SC (24 h., 30 °C, 250 r.p.m.), whereafter, for each replicate, 50,000 cells were transferred to 300 µL volumes comprising a dilution series of respective ligands dissolved in pH-buffered SC-AS/Urea medium[116], and stimulated for 20 h. (30 °C, 250 r.p.m.). DIX45 (Ste2) and DIX59 (No GPCR) was stimulated at pH=5 with 0 µM, 0.01 µM, and 1 µM α-factor (GenScript), DIX60 (Mam2) at pH=5 with 0 µM, 1 µM, and 10 µM P-factor (peptide-seq.: TYADFL-RAYQSWNTFVNPDRPNL) (GenScript), DIX61 (MTNR1A) at pH=5 with 0 µM, 1 µM, and 10 µM melatonin (Sigma-Aldrich, Cat.#M5250), DIX62 (ADORA2B) at pH=7 with 0 µM, 0.01 µM, and 1 µM adenosine (Sigma-Aldrich, Cat.#01890), DIX63 (ADRA2A) at pH=7 with 0 µM, 1 µM, and 10 µM clonidine hydrochloride (adrenaline agonist) (Sigma-Aldrich, Cat.#C7897), DIX64 (5HT4b) at pH=5 with 0 µM, 1 µM, and 10 µM serotonin hydrochloride (Sigma-Aldrich, Cat.#H9523), and DIX65 (CXCR4a) at pH=7 by 0 µM, 100 µM, and 1000 µM NUCC-390 hydrochloride (CXCL12 agonist) (Cayman Chemical Company, Cat.#30957).

For yeast cell staining, approx. 500,000 cells were pelleted (800 g, 3 min., 5 °C), washed twice in 150 µL ice-cold PBSA (1X PBS with 1 g/L bovine serum albumin (Sigma-Aldrich, Cat.#A4503)), then resuspended and incubated in 50 µL primary antibody mix (darkness, 30 min., on ice, 200 r.p.m.). Hereafter, the procedure was repeated for the secondary antibody mix. Finally, cells were pelleted and washed twice in 150 µL ice-cold PBSA, whereafter they were resuspended in 150 µL ice-cold PBSA for flow cytometry using a NovoCyte Quanteon

flow cytometer (Agilent). The CD19 epitope was stained with 2.5 μg/mL (1:200) mouse anti-CD19 (FMC63) (Absolute Antibody, Cat.#Ab00613) and 40 μg/mL (1:50) goat anti-mouse-AF647 (polyclonal) (Thermo Fisher, Cat.#A-21236), detected in R667/30-H, and the HA-tag was stained with 2.5 μg/mL (1:400) rabbit anti-HA (RM305) (Thermo Fisher, Cat.#MA5-27915) and 10 μg/mL (1:200) goat anti-rabbit-AF488 (polyclonal) (Thermo Fisher, Cat.#A-11008), detected in B525/45-H (Fig. 2g–j, Supplementary Fig. 10-12), or alternatively Anti-HA.11-AF647 (BioLegend, Cat.#682404) detected in R667/30-H (Fig. 2a-b, Supplementary Fig. 3). NALM6 was stained analogously for direct comparison of CD19 levels. The gating strategy is described in Supplementary Fig. 32. Each condition was examined in three biological replicates and 50,000 cells were sampled per replicate. A one-way or two-way ANOVA with Tukey's, Dunnett's, or Šídák's multiple comparisons test was conducted for the individual conditions, and to fulfill the assumptions of parametric tests, some data was $log_{10}$-transformed, all specified in Supplementary Data 5-6.

**Yeast growth assays in mammalian media.** For yeast growth assays (Supplementary Fig. 13), strains were seeded at 40,000 cells in 200 μL media and grown at 37 °C for 24 h. in yeast media; SC and pH=5-buffered SC-AS/Urea[116], as well as in human immune cell media; RPMI + 10%FBS + 1%PS and ImmunoCult™-XF T Cell Expansion Medium. Yeast cells were grown in a BioTek Epoch 2 (Agilent) microplate reader to monitor growth at $OD_{630}$ in three biological replicates.

**T cell viability and proliferation in yeast co-cultivations.** For assessing T cell viability and proliferation, as well as yeast biosensing during long-term co-cultivation (Supplementary Fig. 13), T cells separated from a donor-derived isolate of PBMCs were co-cultured with a yeast strain equipped with Ste2 and a $P_{FUS1}$-yEGFP cassette (B525/45-A). For co-cultivation, a constant cell number of 500,000 T cells was supplied with ratios of 0.5x (250,000), 1.0x (500,000), or 10x (5,000,000) yeast cells and cultivated in 500 μL RPMI + 10%FBS, for each replicate, at 37 °C with 5% $CO_2$ for 4 days (96 h). In addition, the 1.0x co-cultures were assessed with supplementation of 20 μM α-factor (GenScript). Yeast was pre-cultivated in 0.5 mL SC (24 h., 30 °C, 250 r.p.m.), then diluted 10-fold and further cultivated in SC (24 h., 30 °C, 250 r.p.m.), and then washed two times in PBS before establishing co-cultures (Supplementary Fig. 31). T cells were labeled using a CellTrace™ CFSE Cell Proliferation Kit (Thermo Fisher, Cat.#C34554) (B525/45-A) according to the manufacturer's protocol before co-culture establishment, and after co-cultivation the entire culture was stained with anti-CD3-BV421 (1:50) (BD Biosciences, Cat.#562426) (V445/45-A), and using a Fixable Near-IR Dead Cell Stain Kit (Thermo Fisher, Cat.#L34975) (R780/60-A) according to the manufacturer's protocol for assessing T cell viability. For staining, the co-cultures were pelleted (300 g, 5 min., 5 °C), then washed with 100 μL Cell Staining Buffer (BioLegend, Cat.#420201), whereafter each replicate was resuspended in 50 μL staining mix and incubated (darkness, 30 min., on ice). Then cells were pelleted (300 g, 5 min., 5 °C), washed with 150 μL ice-cold Cell Staining Buffer, and then resuspended in 150 μL ice-cold Cell Staining Buffer for flow cytometry. The co-cultures were examined in three biological replicates and 50,000 cells were sampled per replicate at day 1 (0 h.), day 2 (24 h.), and day 5 (96 h.) by flow cytometry using a NovoCyte Quanteon flow cytometer (Agilent). The gating strategy is described in Supplementary Fig. 32. A two-way ANOVA with Tukey's multiple comparisons test was conducted for the individual conditions, and to fulfill the assumptions of parametric tests, some data was $log_{10}$-transformed, all specified in Supplementary Data 7.

**CAR Jurkat NFAT-Luc activation assay with SCASA yeast cells and NALM6.** For the investigation of SCASA yeast cell activation of CAR Jurkat NFAT-Luc cells with comparison to NALM6 (Fig. 3, Supplementary Fig. 14), several co-cultures were set up. Yeast strains (DIX44, DIX45,

DIX48, DIX47, and DIX49) were inoculated and pre-cultivated in 0.5 mL SC (24 h., 30 °C, 250 r.p.m.), then diluted 10-fold and further cultivated (24 h., 30 °C, 250 r.p.m.), whereafter for GPCR stimulation, for each replicate, cultures were diluted 50-fold by transfer to 200 μL volumes comprising a 10-fold dilution series of α-factor (0–1 μM) (GenScript) in SC + 1%DMSO, and then stimulated for 20 h. to induce differential CD19 YSD (30 °C, 250 r.p.m.). Before establishment of co-cultures, SCASA yeast cells were pelleted (800 g, 3 min.), washed twice in 150 μL PBS, and resuspended in RPMI + 10%FBS + 1%PS. For the establishment of 0-100% CAR+ Jurkat NFAT-Luc cultures, 0% CAR+ and 100% CAR+ Jurkat NFAT-Luc cultures were titrated by cell counting to control the amount of CAR + cells in each culture. Then, for each replicate of co-cultures, 200,000 alive Jurkat NFAT-Luc cells were co-cultured with 40,000 target cells (i.e. a T/E-ratio of 0.2x SCASA yeast cells or NALM6) for 18 h. in 200 μL RPMI + 10%FBS + 1%PS at 37 °C with 5% $CO_2$ (Supplementary Fig. 31). After co-cultivation, cells were lysed using Pierce™ Luciferase Cell Lysis Buffer (Thermo Fisher, Cat.#16189) and the luciferase activity was examined using the Luciferase Assay System (Promega, Cat.#E1500) according to the manufacturers' protocols. Three replicates were made for each tested condition. Relative activation of Jurkat NFAT-Luc cells was calculated by subtraction of background luminescence and normalization to a 100% CAR+ Jurkat cell mono-culture. A one-way or two-way ANOVA with Tukey's or Dunnett's multiple comparisons test was conducted for the individual conditions, and to fulfill the assumptions of parametric tests, some data was $log_{10}$-transformed, all specified in Supplementary Data 8.

**CAR design screening assay in TPR Jurkat cells using SCASA yeast cells.** To demonstrate the differential impact of target antigen densities and cellular ratios on different CAR designs, a wide variety of co-cultivations were set up (Fig. 4, Supplementary Figs. 15–19). Yeast strains DIX44, DIX45, DIX47, DIX48, DIX49, and DIX67 were inoculated and pre-cultivated in 0.5 mL SC (24 h., 30 °C, 250 r.p.m.), then diluted 10-fold and further cultivated (24 h., 30 °C, 250 r.p.m.). For the differential target-cell-ratio conditions, yeast cultures were pelleted (1,200 g, 90 s.), washed twice with PBS, and then resuspended in RPMI + 10%FBS + 1%PS. Yeast cell densities were then normalized to result in 8.0x−0.25x the number of alive CAR TPR Jurkat cells, by establishing the 8.0x condition through equalizing cell counts, and then serially diluting it to produce 4.0x, 2.0x, 1.0x, 0.5x, and 0.25x yeast cultures. The same was done for NALM6. To establish different antigen-density conditions, each yeast strain was stimulated through the controlling GPCR differentially by 20-fold culture dilutions into a 10-fold dilution series of α-factor (0-1 μM) (GenScript) in full SC. Strains were incubated for 20 h. to induce differential CD19 YSD (30 °C, 250 r.p.m.). After induction of antigen-densities, yeast cells were pelleted (1,200 g, 90 s.), washed twice with PBS, and then resuspended in RPMI + 10%FBS + 1%PS. Then each individual yeast culture, composed of a combination of yeast strain and stimulation level, was normalized to the same cell density to ensure that all CAR TPR Jurkat co-cultures would be exactly 1.0x yeast for all antigen-densities examined. All stimulatory conditions were then set up by combining the target cells with the CD28 CAR and 4-1BB CAR TPR Jurkat cell lines individually, always with 50,000 alive CAR TPR Jurkats cells in a total volume of 200 μL RPMI + 10%FBS + 1%P/S (1.0x). In addition to the co-cultures, negative control monocultures and positive control cultures with 1.0x Human T-Activator CD3/CD28 Dynabeads (Thermo Scientific, Cat.#11161D) (αCD3/αCD28) were employed. The cells were then co-cultured for 24 h. at 37 °C with 5% $CO_2$. Finally, cells were pelleted (300 g, 5 min.) and resuspended in 150 μL ice-cold PBSA for flow cytometry using a NovoCyte Quanteon flow cytometer (Agilent); NF-κB-CFP (V525/45-A), NFAT-eGFP (B525/45-A), and AP-1-mCherry (Y615/20-A). Three biological replicates were made for each condition. Responsiveness analysis was done by $log_{10}$-transforming response data and conducting linear regressions (Supplementary Data 11). Two-

way ANOVA with Šidák or Tukey's multiple comparisons test was conducted for the individual analyses, and to fulfill the assumptions of parametric tests, data was $log_{10}$-transformed when relevant, all of which is specified in Supplementary Data 10-11.

**Comparison of SCASA yeast cells to microbeads and microtiter plates.** To benchmark the performance of SCASA yeast cells against commonly used non-cellular CAR T cell activation methods, we employed antigen-coated microbeads and microtiter plates (Fig. 4, Supplementary Figs. 20–24). Specifically, purified biotinylated human CD19-AviTag™ (ACROBiosystems, Cat.# CD9-H82E9) was attached either to 5.5 μm paramagnetic ActiveMax® streptavidin-coupled μbeads (microbeads) (ACROBiosystems, Cat.# MBS-C009) or to 96-well streptavidin-coated flat-bottom microtiter plates (ACROBiosystems, Cat.#SP-15). CD19 was resuspended and stored as 200 μg/mL aliquots in 0.2 μm-filtered PBS with 10% trehalose (pH=7.4) at −80 °C, microbeads were resuspended and stored as 5 mg/mL aliquots ($50 \times 10^6$ beads/mL) in 0.2-filtered PBS with 0.1% HSA and trehalose (pH=7.4) at −80 °C, and microtiter plates were stored at 4 °C, all according to manufacturers protocols. CD19 protein was thawed on ice, and a 2-fold dilution series was prepared using sterile PBS to enable control of antigen density gradients on microbeads and microtiter plates. Microbeads were thawed on ice, after which they were washed twice with excess ice-cold sterile PBS to remove the storage buffer. Gentle mixing by pipetting was performed during each wash, followed by magnetic separation using a Magnetic Stand-96 (Thermo Scientific, Cat.#AM10027). Microbeads where then resuspended in aliquots of sterile PBS at $50 \times 10^6$ beads/mL to be combined with the CD19 dilutions. Specifically, for each antigen density, 40 μL of microbeads ($2 \times 10^6$ beads) was combined with 40 μL of 2X CD19 dilution ($25 \times 10^6$ beads/mL) and gently mixed in 96-well V-bottom plates, followed by 60 min. incubation at room temperature. 100 μL sterile PBSA was then added to each reaction and the microbeads were pelleted (300 g, 5 min.). The CD19-coupled microbeads were then washed three times in sterile PBSA to remove unbound CD19 by pelleting the microbeads (300 g, 5 min.). Finally, the microbeads were resuspended in RPMI + 10%FBS + 1%PS for use in CAR TPR Jurkat cultivations. For the CD19-coating of microtiter plates, 50 μL of each 1X CD19 dilution was added to the wells, followed by 60 min. incubation at room temperature and then three washes with sterile PBSA by pipetting.

To assess microbead loading capacity and resulting antigen densities, a 2-fold dilution series of CD19 was prepared, starting with an excess concentration of 100 μg/mL (2 μg total) to saturate the microbeads. A Quantitative Analysis KiCD3t (QIFIKIT) (Agilent, Cat.#K0078) and 5 μg/mL mouse anti-CD19 (FMC63) (Absolute Antibody, Cat.#Ab00613) was then employed to quantify the number of CD19 molecules per microbead by flow cytometric measurement of FITC (B525/45-A) using a NovoCyte Quanteon flow cytometer (Agilent), all according to manufacturers' protocols (Supplementary Fig. 20). In parallel, the CD19 antigen densities of SCASA yeast strains were quantified using the same method; $P_{PGK1}$-Empty (DIX44), $P_{FUS1}$-CD19 (DIX45), $P_{MFA2}$-CD19 (DIX48), $P_{PGK1}$-CD19 (DIX47), and $P_{TDH3}$-CD19 (DIX49) (Supplementary Fig. 24). For the CD28 CAR and 4-1BB CAR TPR Jurkat experiments, each replicate contained 200,000 alive CAR TPR Jurkats cells in a total volume of 200 μL RPMI + 10%FBS + 1%P/S. Microbeads and SCASA yeast were added to maintain identical T/E-ratios with CAR TPR Jurkats for all conditions (1.5x the number of alive CAR TPR Jurkat cells). For plate-bound CD19, CAR TPR Jurkat cells were directly added to the coated microtiter plates. Cells were cultured for 24 h. at 37 °C with 5% $CO_2$. Prior to yeast co-cultivation, yeast strains DIX44, DIX45, DIX47, DIX48, and DIX49 were inoculated and pre-cultivated in 0.5 mL SC (24 h., 30 °C, 250 r.p.m.), then diluted 10-fold and further cultivated (24 h., 30 °C, 250 r.p.m.). Yeast cultures were pelleted (1200 g, 90 s.), washed twice with sterile PBS, resuspended in RPMI + 10%FBS + 1%PS, and normalized to the same cell density, as

described above for other co-cultivation experiments. CAR TPR Jurkat activation was characterized by flow cytometry and data analysis, as described in the above sections (Supplementary Data 12-13).

**Activation assay of donor-derived CAR T cells with SCASA yeast cells and NALM6.** SCASA yeast cell activation of donor-derived CAR T cells was done by co-cultivation with comparison to NALM6 (Fig. 5, Supplementary Figs. 25–28). Yeast strains $P_{PGK1}$-Empty (DIX44) and $P_{PGK1}$-CD19 (DIX47) were inoculated and pre-cultivated in 0.5 mL SC (24 h., 30 °C, 250 r.p.m.), then diluted 10-fold and further cultivated (24 h., 30 °C, 250 r.p.m.). Before the establishment of co-cultures, SCASA yeast cells were pelleted (800 g, 3 min.), washed twice in 150 μL PBS, and resuspended in RPMI + 10%FBS + 1%PS. For co-cultivation, a constant cell number of 100,000 alive CAR+ T cells was supplied with ratios of 0.2x (20,000), 1.0x (100,000), or 5.0x (500,000) SCASA yeast cells or NALM6 cells and cultivated in 200 μL RPMI + 10%FBS + 1%PS for each replicate. For CTRL T cell co-cultures, the corresponding number of alive T cells was used to maintain the same overall number of T cells. The co-cultures were incubated for 20 h. at 37 °C with 5% $CO_2$ (Supplementary Fig. 31). After co-cultivation, the cultures were stained for flow cytometry using anti-CD69-PE/Cy7 (1:150) (BioLegend, Cat.#310912) (Y780/60-A), anti-CD3-perCP (1:30) (BioLegend, Cat.#300326) (B690/50-A), anti-c-myc-DL488 (1:40) (Abcam, Cat.#ab117499) (B525/40-A), anti-CD19-BV785 (1:50) (BioLegend, Cat.#302240) (V780/60-A), and the Zombie Violet Fixable Viability Kit (BioLegend, Cat.#423114) (V445/45-A) according to the manufacturer's protocol. A master mix was made in Cell Staining Buffer (BioLegend, Cat.#420201). For staining, the co-cultures were first pelleted (300 g, 5 min., 5 °C) to then carefully aspirate the media, whereafter each replicate was resuspended in 50 μL staining master mix and incubated (darkness, 30 min., on ice). Hereafter, the cells were pelleted (300 g, 5 min., 5 °C), washed twice with 150 μL ice-cold Cell Staining Buffer, and then resuspended in 100 μL ice-cold Cell Staining Buffer for flow cytometry. Three biological replicates were made for each condition, and measurements were conducted using a CytoFLEX-S (V-B-Y-R) instrument (Beckman Coulter) by analyzing 80 μL of each replicate. The gating strategies are described in Supplementary Fig. 33. A one-way, two-way, or three-way ANOVA with Tukey's multiple comparisons test was conducted for the individual conditions, and to fulfill the assumptions of parametric tests, some data was $log_{10}$-transformed, all specified in Supplementary Data 14. The performance summary of the investigated conditions (Fig. 5i) shows the mean performance across all T/E ratios (0.2x, 1.0x, 5.0x) for SCASA yeast $P_{PGK1}$-CD19 and NALM6, for each parameter displayed in the individual plots (Fig. 5a-h). This was done by calculating the mean of all measurements normalized to the global maximum value detected, standardized to 100% for each parameter, regardless of target cell type and T/E ratio.

After 5 months of cryopreservation in liquid nitrogen, the CAR T cell product was thawed for reexamination (Supplementary Fig. 25). After thawing, the CAR T cells were recovered by two days of cultivation in ImmunoCult™-XF T Cell Expansion Medium (Stemcell Technologies, Cat.#10981) supplemented with Human Recombinant IL-2 (12.5 ng/mL) (Stemcell Technologies, Cat.#78036), with initial seeding at $1 \times 10^6$ alive cells/mL. Yeast strains $P_{PGK1}$-Empty (DIX44), $P_{MFA2}$-CD19 (DIX48), and $P_{TDH3}$-CD19 (DIX49) were pre-cultivated with and without stimulation using 0.1 μM α-factor (20 h., 30 °C, 250 r.p.m.), as described for CD19 display characterization. SCASA yeast cells were pelleted (800 g, 3 min.), washed twice in 150 μL PBS, and resuspended in RPMI + 10%FBS + 1%PS. Co-cultures were then established by combining 75,000 alive CAR+ T cells with 25,000 yeast cells (0.3x) in 200 μL RPMI + 10%FBS + 1%PS. A positive activation control was made using 1X Leukocyte Activation Cocktail (LAC), with BD GolgiPlug™ (BD Biosciences, Cat.#550583) as positive control. After 20 h. of cultivation at 37 °C with 5% $CO_2$, cultures were stained for flow cytometry with anti-CD69-PE/Cy7 (1:150) (BioLegend, Cat.#310912) (Y780/60-A), anti-CD25-BB700 (BD Biosciences, Cat.#566448) (B695/40-A), anti-CD3-

FITC (BD Biosciences, Cat.#349201) (B525/45-A), and the Zombie Violet Fixable Viability Kit (V445/45-A), as described above. The gating strategies are described in Supplementary Fig. 33. Two biological replicates were made for each condition, and measurements were conducted using a NovoCyte Quanteon flow cytometer (Agilent).

## Software

For the collection of flow cytometry data, NovoExpress Software v1.6.2 (Agilent) was employed for controlling the NovoCyte Quanteon system (Agilent) and CytExpert Acquisition and Analysis Software v2.5 (Beckman Coulter) for the CytoFLEX-S system (Beckman Coulter). Flow cytometry data was analyzed using FlowJo™ v10.8 Software (BD Life Sciences). Statistical analyses and plotting were done using GraphPad Prism v9.4.1 (GraphPad Software). Schematic figure illustrations for Figs. 1, 2c, 3a, 4a, and Supplementary Fig. 30, 31, were created with BioRender.com.

## Reporting summary

Further information on research design is available in the Nature Portfolio Reporting Summary linked to this article.

## Data availability

Source data and analyses generated for this study are provided with this paper as a Source Data file and as a Supplementary Data file, respectively. Source data are provided with this paper.

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

## Acknowledgements

This work was supported by the Novo Nordisk Foundation, grant number NNF20CC0035580 to M.K.J., grant NNF21OC0066562 "Challenge Programme 2021 - Smart Nanomaterials for Applications in Life-Science" to S.R.H., the U.S. Department of Energy, grant number DE-SC0018368 to R.T.G., the Technical University of Denmark, DTU Skylab Discovery

Grant to M.D. and E.D.J., and the Independent Research Fund Denmark, grant number 0129-00005B to M.O. Authors would like to thank Dr. Jie Zhang for providing plasmids pESC-URA-gRNA_XI-3 and pESC-URA-gRNA_XII-5, as well as Professor Tom Ellis and Dr. William Shaw for plasmid pWS1776.

## Author contributions

M.D., E.D.J., S.R.H., M.O., and M.K.J. conceived the study. M.D. led the investigation. M.D. performed yeast characterization experiments with support from N.M.T.K. M.D. designed and constructed all yeast strains and plasmids of this study, except for DIX32-34 and pMAD26,29-30 constructed by N.M.T.K. M.D., K.R., and G.S. performed Jurkat co-cultivation experiments. R.U.F.W., K.R., and M.O. constructed the CAR Jurkat NFAT-Luc cell line and CAR TPR Jurkat cell lines. K.Z., M.D., and G.S. constructed donor-derived CAR T cells and performed co-cultivation experiments. M.D. performed all data analyses. M.D., E.D.J., and MKJ wrote the manuscript. M.K.J., M.D., E.D.J., R.T.G., S.R.H., and M.O. acquired the funding enabling this research. All authors approved the manuscript.

## Competing interests

The authors declare the following competing interests: M.D., M.K.J., E.D.J., and G.S. are named as inventors on a pending patent application related to the SCASA technology. The remaining authors declare no competing interests.
