## [Transparent Peer Review file · Nature Communications]

A Yeast Surface Display Platform for Characterizing CAR T Cell Responses to Cancer Antigens

Corresponding Author: Dr Michael Jensen

Version 0:

Reviewer comments:

Reviewer #1

(Remarks to the Author)

In this manuscript, the authors present a yeast-based system that allows controllable surface display of cancer antigens, using the antigen CD19 as a model. By co-cultivating the engineered yeast strains with CAR-T cells, this allows controlled activation of CAR-T cells to assess response characteristics, such as antigen-density thresholds. More specifically, the yeast-based system, called Synthetic Cellular Advanced Signal Adapter (SCASA), combines yeast surface display (YSD) technology with a refactored GPCR sensing system in yeast designed to accept heterologous GPCRs (PMID 30955892). Together, this allows tunable expression of displayed antigens on yeast cells in response to extracellular GPCR ligand inputs. The authors demonstrate that their system enables tunable regulation of CD19 display. Furthermore, using the SCASA yeast system, they demonstrate activation of CAR Jurkat cells (encoding a NFAT reporter) as well as primary CAR-T cells, comparing their yeast-based antigen display system against conventional CD19+ NALM cells.

This work addresses an important problem: the need for methods to study the relationship between antigen density and CAR-T activation. Currently, there is little known about what influences the threshold for antigen recognition, and we lack solutions to address the problem of cancer cell heterogeneity. The method proposed – to use a highly customizable and controllable yeast-based system to present antigen profiles to CAR-T cells – is clever, interesting, and potentially quite powerful. Thus, I believe this manuscript can be a timely and compelling contribution, well-suited to the readership of Nature Communications.

The manuscript represents high-quality work, and includes novel elements that should be useful to the field. The paper is well written and relevant papers are referenced. Overall, I believe there are valuable aspects of this work and that there is a strong paper here, but these are hard to uncover in the current presentation of the manuscript. In its current form, I find aspects of the manuscript confusing. Principally, the work appears to have multiple aims which do not necessarily fit together (a tunable GPCR signaling toolkit and a new yeast display method for CAR-T cell activation). In principle, I can see some advantages of combining these aims, the current manuscript doesn't in my opinion provide a convincing enough argument.

As mentioned above, the broad goal of creating a platform to study the relationship between antigen density and CAR-T activation is compelling and important. I recommend drastically simplifying and streamlining this manuscript to achieve this, which is a yeast display method for CAR T cell activation. However, this would also need to be better compared to or discussed in the context of other recent similar work: PMID 35379810 (yeast display for antigen density studies) and PMID 32193224 (library of NALM clones with a variety of CD19 densities to evaluate CAR designs).

Below are critiques with suggestions on how to streamline the paper to benefit the intended audience, and necessary points of discussion to properly highlight the pros and cons of this work. Addressing these points will, in my humble opinion, enhance the manuscript and make it suitable for publication in Nature Communications.

MAJOR COMMENTS

First and foremost, too much data is being presented in this manuscript. A lot of this work has been demonstrated before or is unnecessary to the main story. This should be streamlined to properly highlight the novel and useful aspects of this work.

It is unclear why GPCRs are being used for this system over a much simpler inducible promoter system or a panel of constitutive promoters with varying strengths. I recommend the authors provide stronger rationale. Choosing this (more complicated) approach is further complicated by the stated motivation of having an orthogonal yeast system which is induced by a highly specific fungal peptide, which is then substituted for human GPCRs which are induced by human relevant ligands. The use of mammalian receptors introduces potential complications, such as crosstalk between yeast and T cell GPCR induction, which is not addressed. Additionally, the alternative GPCRs do not seem to be beyond the initial demonstration that they can tune the output.

Thus, my recommendation would be to remove the mammalian receptors entirely, focus on Ste2, and better highlight the benefits of the PRP system over inducible systems and constitutive promoters. One of the advantages of the PRP shown in this work is the additional expression boost which occurs due to morphological changes in the yeast, which is a well-known phenomenon (documented in Ref 53 Fig S2b, for example), and likely due to larger cells. However, it is not demonstrated whether this represents a higher CD19 density on the yeast cell surface or the yeast has a larger cell surface (and therefore more CD19), as indicated by the SSC normalization which mostly subtracts these differences. FSC normalization should also be performed here to better assess whether this is a size effect.

It is therefore unclear what is the key factor for CAR T stimulation, CD19 density per cell or total CD19 in the co-culture? This is crucial for understanding the best way for other researchers to take this system and apply it to CAR T cell activation. Can I just pick a single constitutive promoter and vary the density with cell count, as shown in Fig. 5? If so, is the inducibility aspect of this work necessary? It seems all of the data is here to better convey this message, but it needs to be presented in a clear manner. Additionally, the pros and cons of the system need to be discussed in far more detail.

Characterizing pheromone responsive promoters is fairly well-established and thus does not require a dedicated figure. Perhaps combining Figs 2+3 to show CD19 display can be tuned using PRP promoters – this is the novel part of the work. Inverse promoters are also not relevant to this work.

The Pgal1 system was optimized by integrating the construct into the genome. The rationale for not using this commonly used inducible promoter was a digital response. However, it is never contrasted with the GPCR system, only a constitutive promoter. Furthermore, Pgal1 can be converted into an analogue response with a single deletion of GAL2, which is not provided as a simple solution to the binary behavior. Arguably this would be far simpler than the multiple gene KO's required for using the PRP. Again, the motivation for not using a standard inducible promoter (or constitutive promoter) here is weak. Additionally, the improved integrated galactose inducible system should be the comparison in Fig3b, not the conventional EBY100 plasmid-based system, as this is instead a comparison of integration vs episomal, which is not the argument being made.

In terms of the SCASA system being applied to study CAR activation, I recommend bolstering the discussion/rationale for why this approach may provide unique benefits vs. conventional NALM (or other cancer) cell lines engineered with different antigen levels. This is partly addressed in the observations in Figure 5 and in the discussion; 1) there is "CAR-dependent NALM6 antigen-down modulation" and 2) "cytotoxic sensitivity of NALM6 imposing a selective pressure bias and loss of CD19+ cells". However, from a translational point of view, it still may be preferable to work with human cancer cells when evaluating different CAR designs to connect design, antigen density and desired phenotype (i.e. cell death which cannot be assessed with yeast). If a benefit of the SCASA system is to enable a more robust and high-throughput platform than can be achieved when using human cancer cells, this should be demonstrated or, at the least, further explained.

While not strictly essential, it would certainly enhance the paper to test >1 CD19 densities with primary T cells (Figure 5), to demonstrate CAR activity with a variety of CD19 densities. Only one yeast CD19 density (PPGK1) was implemented in co-culture with primary cells while several were quantified with Jurkat cells (Figure 4). This is partially addressed in supp figure 11 with the addition of two promoters (MFA2 and TDH3), however I am unsure what alpha factor was used in these figures and am concerned that there is not a significant difference in CD25 expression between induced and uninduced yeast expressing CD19. Overall, I think there is a missed opportunity to use these demonstrations with Jurkat and primary T cells to showcase in what settings and/or applications SCASA is beneficial, superior, or powerfully complementary to using human cell lines.

It should be explained why these two CAR constructs (FMC63-CD8.-4-1BB-CD3 and Hu19-CD8.-CD28-CD3) were chosen and why there was a change when going from Jurkats to primary. This is important since it has been indicated in citation 38 and referred to in line 388 that ICD CD-28 outperforms 4-1BB in relatively low antigen densities. It would have been great if it could have been demonstrated using SCASA that these CARs produce different T cell activation levels when varying antigen density.

MINOR COMMENTS

If PRP is being stimulated AGA1 might be expressed at higher levels, which may also explain the boost in CD19 display. This was not discussed but could be an advantage of PRP induction for yeast display purposes?

Line 83: Please expand on what you mean by limited control of antigen densities.

Fig 4b: what are the cell ratios being used here? Why were these 4 promoters chosen vs the 8 tested in 3E? Please make it more clear what the threshold level is for CAR activation - is it when a condition is significantly different than 0uM of alpha factor?

Figure 4c: How were %CAR+ Jurkat populations made?

Line 192: It is not surprising HPF1 and CSS1 are downregulated as the mRNA data already showed this. Again, inverse promoters are not relevant to this work.

Line 225: This is not sufficient evidence to say conventional YSD methods are unfit for high-precision cellular communication. Or if it is, you could comment on small improvements which could improve them, outside of the GPCR approach.

Line 378: What is meant by 100% CAR+ Jurkat NFAT-Luc cells. What control is this compared to?

Figure 5a: What concentration of alpha factor was used?

Line 544: inverse agonism of PRP is not a novel insight (ref 60) and PRP-induced expression boost has previously been shown (ref 53).

Line 566: Please expand on this: "Hence, we specifically regard the SCASA yeast platform as useful for applications where specific signals must be isolated from the complex interaction network"

Reviewer #2

(Remarks to the Author)

Deichman et al. engineered a yeast-based model to activate CAR T cells with a tunable level of the CD19 antigen. The authors used GPCRs and a-factor to induce a chimeric protein expressing the CD19 EC domain. Yeast expressing different levels of CD19 were then cultured with human T cells, benchmarking them against T cells activated with the validated leukemia cell line NALM6. NFAT activation in the Jurkat reporter system and CD69 upregulation were measured. The design presented by the authors is elegant, and the manuscript is well-presented. However, the system presented here is artificial and lacks the critical dynamics of T cell/tumor cell interplay. There is a notable difference in the levels of CD19 between NALM6 and the yeast system, and the degree of T cell activation varies significantly across these models. As acknowledged by the authors, the yeast system presents considerable differences compared to standard cell lines. It's easy to engineer tumor cells with a gradient of antigens using sh-RNA, CRISPR-Cas9-KO, inducible systems, or viral vectors in a lab, while the yeast system might be challenging to implement. Natural targets are not expressed in the yeast system, thus raising concerns whether these findings can be generalized. Moreover, long-term mechanistic studies are lacking.

Reviewer #3

(Remarks to the Author)

Deichmann et al. present a yeast surface display-based platform for the graded presentation of CD19 as a tumor-associated antigen to chimeric antigen receptor-modified Jurkat cells and primary T-cells (CAR-T-cells). Using a GPCR for signal input as well as engineered Pheromone Response Pathways for further processing and Aga1/2 as a yeast cell wall anchor, the authors achieve titratable expression of a thermostable mutant CD19 on the yeast surface. The study assesses the yeast's suitability as an antigen-presenting cells (APC)/target cell by confronting CAR-expressing Jurkat cells and primary T-cells with such yeast cells as well as with CD19-positive NALM6 cells. NFAT transcriptional activity, CD69 activation marker and CD19 CAR downregulation are employed as readouts for CAR-T cell stimulation.

As a synthetic biologist, I commend the authors for establishing this elegant yeast-based platform, testing it on T-cells, and employing a quantitative approach toward T-cell responses. Furthermore, the study benefits from a comprehensive language and clear structure. However, from the perspective of an immunologist and T-cell biologist, my overall enthusiasm is limited.

Major points:

(i) Use of yeast as an APC

While the presented yeast system offers some degree of "dialability" (please also refer to my comment below), it comes with a number of ambiguities related to the biology of the CAR-T-cell-APC interaction. (a) The authors are right about mentioning them in the discussion section, but underestimate the consequences resulting from co-incubation, some of which may not even been that well studied. I noticed that the co-culturing was limited to 20 hours, probably for good reasons, as yeast grows faster than T-cells, competes for nutrients and contaminates the media with yeast-related metabolites.

(b) There is a lot to be said about the geometry of membrane apposition, with the dimensions of the intermembrane space and the lengths of receptor-ligand pairs regarded as a critical factor for kinetic segregation of kinases and phosphatases (and hence CAR-proximal signaling). In this regard, the yeast system with the cell wall serving as anchor for CD19 (or other TAAs), is poorly defined. I think there are more defined and equally available/accessible platforms that are up to the task of ligand scalability (see (iii) below).

(ii) Broad range of expression

Figure 3D illustrates the ranges of scalable CD19 expression in various yeast systems. The dynamic range is limited and the range of surface expression is rather broad, also for the P_PGK1-CD19 variant, which is employed in experiments involving primary human CAR-T-cells. Such broad expression and overlapping expression levels (due to a limited range in

expression) results in undesirable noise, which may render a detailed analysis of the heterogeneity in the response of any given CAR-T-cell population challenging if not impossible.

(iii) Other platforms outperforming the yeast system

There are other platforms available which allow for the titratable presentation of TAAs to CAR-T-cells. (a) Electroporation of target cells with capped in vitro-transcribed mRNA (PMID: 32820173) gives rise to narrow ranges of expression (for each mRNA concentration, e.g. in Jurkat cells) with a highly dynamic range of expression (depending on the amount of mRNA added to the electroporation). Advantage: highly titratable with narrow expression.

(b) CombiCells (PMID: 38177315): The use of the Spy-catcher system allows for narrow and highly titratable surface display of recombinant antigens as well as accessory factors.

(c) (Strept-)Avidin-based solid platforms (e.g. beads or planar surfaces) allows for highly titratable and quantifiable surface display of recombinant antigen and accessory molecules. It is highly scalable and adequate for high-throughput analysis.

(d) Protein-functionalized planar glass-supported lipid bilayers (PMID: 32632291): advantages are as described in (c) but the system also allows for dialing in the lateral mobility of membrane associated recombinant antigens and accessory factors while it is still compatible with medium-throughput analysis.

All platforms are highly quantitative and support co-culture periods longer than 72 hours. (b) allows for killing experiments, while (c) and (d) still afford quantitation of degranulation (with the use of an anti-lamp-1 antibody).

Taken together, I did not perceive any advantages of the yeast-based system in the special context of cell-cell recognition that would make me choose it. Rather the opposite is true. Yeast is a fungus triggering TLR-responses in T-cells (as discussed by the authors) and with a cell anatomy that is considerably smaller than that of T-cells. This leaves an open question as to whether a given CAR-T-cell is associated with one, two or more yeast cells at any given time (e.g. at an E:T ratio of 1:5) further complicating the analysis.

In conclusion, I see interesting aspects of the presented study within the area of synthetic biology, yet I find it difficult to find additional value within the yeast system for the study of cell: cell interactions, as there are already more defined and highly accessible systems available to the community that support population / single cell as well as high throughput analyses.

Version 2:

Reviewer comments:

Reviewer #1

(Remarks to the Author)

In their revision and rebuttal, the authors have satisfactorily addressed many of the concerns raised by me and the other reviewers. In particular, they have:

- Streamlined the presentation of their system, data, and manuscript
- Better outlined the pros and cons of the SCASA system vs. existing antigen presentation platforms (e.g. microbeads, microtiter plates, cancer cell lines, etc.), including through benchmarking experiments.
- Included a new experiment in which different CAR designs and Jurkat activation reporters are tested against 50 different yeast-based stimulatory conditions, thus illustrating the potential of their system to facilitate high-throughput experiments.

Together, these revisions help to highlight the unique capabilities of SCASA relative to other systems and when it may make sense to use SCASA. Lingering questions will remain about the merits of using a yeast-based system for these types of studies (as noted by Reviewer #3), including questions about the nature of how antigens are being presented, but overall I think the authors have convincingly shown that their genetic and tunable system is indeed capable of stimulating and screening CAR designs in a manner that makes it applicable to future studies.

Reviewer #2

(Remarks to the Author)

Reviewer #3

(Remarks to the Author)

In its current revised version of the ms. the authors have gone to great lengths to address the concerns I felt necessary to raise in my evaluation of their first submission. While I continue to wonder about the need to implement a yeast-based system as a means to functionally validate CAR-T-cell, I am more than open to making the public aware of it for the sake of much needed progress. The use of yeast may in fact fertilize the discussion concerning CAR-T and TCR-T screens, e.g. aimed at TCR-deorphanization or improving CAR-function.

Since all my conceptual concerns related to the use of yeast have been thoroughly and satisfactorily addressed, I consider the revised version adequate for publication in Nature Communication.

Response to Reviewers - Deichmann et al. 2025

We thank the reviewers for the positive comments on our work, for recognizing the potential and strengths of our platform, and for the detailed review with great attention to detail. The specific and constructive feedback has been instrumental in helping us plan and execute new benchmark experiments and also simplify the manuscript. From this major revision, we have obtained exciting results that clarify the value proposition of SCASA to the reviewers. Thank you once again for your valuable contributions. Please find our point-by-point response to each comment below.

Response to reviewer #1:

1. Simplification and streamlining of data presentation:

“First and foremost, too much data is being presented in this manuscript. A lot of this work has been demonstrated before or is unnecessary to the main story. This should be streamlined to properly highlight the novel and useful aspects of this work.”

We agree with the reviewer that a major simplification and streamlining of the story was needed to improve the manuscript. Hence, we have conducted a major re-writing and re-structuring of the result sections, especially to remove text sections that covered the building of the yeast platform:

a) **Condensing and re-structuring result sections:**

Large proportions of text have been removed from the result sections found in the first submission by condensing or removing excessive details, especially yeast engineering already known in the literature.: 1,050 words removed.

The result sections on yeast engineering have been heavily re-structured into new more logical sections in a new order;

- i) Intro to SCASA: ***“Designing the Synthetic Cellular Advanced Signal Adapter (SCASA) system”***
- ii) Yeast-surface display: ***“CD19 yeast surface display for cancer cell simulation” (new)***
- iii) GPCRs: ***“Sensory modules using heterologous GPCRs allow for customizable input” (new)***
- iv) Pathway engineering: ***“An engineered pheromone response pathway enables a customizable processing module” (new)***

This has been done through the following actions to the original sections:

- ***“Designing a cancer-simulating yeast cell”*** (reduced by 180 words) - this section comprised a lot of argumentation for the designs, and has instead been condensed to a section that introduces the

reader to the priorities of the SCASA design and primes them for the following engineering sections (*“Designing the Synthetic Cellular Advanced Signal Adapter (SCASA) system”*).

- ***“Characterization of SCASA processing modules”*** (removed) - this section has been greatly reduced into a single paragraph in the results for pathway engineering, conserving the main conclusions specific to our findings; dynamic range and impact of promoter choice, as well as PRP-boost dynamics, and is a part of the new result section *“An engineered pheromone response pathway enables a customizable processing module”*. We have removed the result specifications that reproduced analogous knowledge to current yeast engineering literature (i.e. promoter characterization using yEGFP).
- ***“Modules for dynamic regulation of CD19 yeast surface display”*** (removed) - this section covered a broad range of information - and the main findings have been divided into *“CD19 yeast surface display for cancer cell simulation”* for describing YSD strategies in the beginning of the results, and *“An engineered pheromone response pathway enables a customizable processing module”* for the dynamic control of CD19 levels after the description of GPCR engineering. We have condensed design argumentations, removed excessive descriptions of each promoter, and edited descriptions of results on P_{GAL1}-induction.
- ***“Coupling of heterologous GPCRs allows for a customized sensory module”*** (edited) - this section has been reduced to describe customization of input and edited to enforce the rationale of using GPCRs, and now comprises the new second result section *“Sensory modules using heterologous GPCRs allow for customizable input”*.

The result sections employing human cells are less modified, but with the following changes:

- ***“Co-cultivation of yeast and human T cells”*** (edited) - maintained with slight re-wording to improve clarity.
- ***“Controlled human immune cell activation by SCASA yeast cells”*** (edited) - condensed by removal of excessive details for the individual yeast designs and CAR Jurkat NFAT-Luc responses, but maintaining information supporting the main conclusions.
- ***“SCASA yeast applications in characterizing CAR designs and downstream signaling”*** (new) - this section is completely new, and is based on new experimental work to address the reviewers' comments.
- ***“Characterizing a donor-derived CAR T cell product using SCASA yeast cells”*** (edited) - this section is largely maintained, however, condensed to convey the main message: SCASA yeast could confirm the functionality of a novel CAR T cell product.
- ***“SCASA yeast cells are efficient and robust CAR T cell activators”*** (edited) - maintained with slight re-wording and definitions to improve clarity.

- b) **Combining Fig. 2 and Fig. 3:** In the streamlining of the initial result section on yeast engineering, we have also combined the previous Fig. 2 and Fig. 3, into a new Fig. 2, as recommended by the reviewer (*see details for changes below*).
- *Figure 2:* Fig. 2b,2d,2e have been conserved in the new figure, while Fig. 2a has been removed, and Fig. 2c,2f have been moved to the supplementary.
 - *Figure 3:* Fig. 3a has been updated to contain details from Fig. 2a, while Fig. 3b,3c,3f have been moved to the supplementary. Fig. 3f has been made into a heatmap to align design styles across yEGFP and CD19 data. In addition, Fig.3d and Fig.3e have been combined to form a bar plot with associated histograms.
- c) **Changing Fig. 5:** Fig. 5 has been restructured and provided with a central performance summary panel (Fig. 5i), which summarizes the main point of each of the subpanels to provide a better overview of our key findings from comparing NALM6 cancer cell lines to SCASA yeast cells.

2. Clarifying the rationale for the use of the GPCR platform and PRP engineering

“It is unclear why GPCRs are being used for this system over a much simpler inducible promoter system or a panel of constitutive promoters with varying strengths. I recommend the authors provide stronger rationale. Choosing this (more complicated) approach is further complicated by the stated motivation of having an orthogonal yeast system which is induced by a highly specific fungal peptide, which is then substituted for human GPCRs which are induced by human relevant ligands. The use of mammalian receptors introduces potential complications, such as crosstalk between yeast and T cell GPCR induction, which is not addressed. Additionally, the alternative GPCRs do not seem to be beyond the initial demonstration that they can tune the output.”[...]“Thus, my recommendation would be to remove the mammalian receptors entirely, focus on Ste2, and better highlight the benefits of the PRP system over inducible systems and constitutive promoters.”

We thank the reviewer for highlighting the importance of a clear rationale as to the choice of inducible system. We have enhanced the focus on describing features of the GPCR system in the results and discussion.

- We agree that other inducible systems could allow for antigen-density modulation, such as shown for galactose-induction and as imaginable for transcription factor-based systems, however, there is a set of unique features that made us choose GPCRs:
 - extracellular sensing with intracellular signal transduction,
 - choice of response processing to the same input signal - exemplified by differential regulation obtained from the use of different processing module promoters (e.g. P_{FUS1} and P_{MFA2}),
 - highly sensitive concentration-dependent analog signal transduction,
 - high dynamic output range,
 - high engineerability to allow customization, characterized by modularity and tunability,
 - future-proofing of the SCASA system for engineering opportunities, such as multiplexing (i.e. more expression cassettes) and feedback systems (e.g. responsiveness to T cell outputs),
 - less requirements for the yeast growth media - e.g. glucose is allowed, unlike for gal-based systems that require a metabolic shift in carbon sources,
 - wide variety in choice of input signal - e.g. small-molecules, peptides, and proteins - and user-manageable operational ranges of these to provide system control.

To our knowledge, no other inducible yeast system currently allows this combination of features. These advantages are referred to throughout the introduction, result section, and discussion, where relevant. In relation to this matter, see also our answer specific to gal-systems below (*pt. 6*).

- We agree with the reviewer that the GPCR system is more complicated to engineer from the get-go. However, this also provides more tunability (*see list of advantages above*), and as can be seen from the

P_{MFA2} design characterized in this first SCASA study, the system already now offers a single entry point for tuning antigen expression to elicit a full range of physiologically-relevant T cell responses. All designs are of course available to the scientific community for further exploration and exploitation.

- We acknowledge the recommendation to limit the scope to Ste2. However, for readers not familiar with GPCR platforms in yeast, we have chosen to keep all GPCRs in the paper to demonstrate diverse antigen expression control in yeast for different inputs.
- We have now addressed the exchange of GPCRs and crosstalk in the discussion. Our approach is to test a broad panel of GPCRs to uncover good designs and showcase customizability (e.g., to inspire potential feedback systems). We then deliberately chose the best-performing design (Ste2), which is also an orthogonal GPCR to T cell physiology that we agree is optimal for the CAR T cell applications showcased in this study.
- We thank the reviewer for pointing out the needed structuring of this argumentation and hope that the reviewer finds the presentation of GPCRs in the revised paper sufficiently clarified.

3. Points on the PRP boost mechanism

“One of the advantages of the PRP shown in this work is the additional expression boost which occurs due to morphological changes in the yeast, which is a well-known phenomenon (documented in Ref 53 Fig S2b, for example), and likely due to larger cells. However, it is not demonstrated whether this represents a higher CD19 density on the yeast cell surface or the yeast has a larger cell surface (and therefore more CD19), as indicated by the SSC normalization which mostly subtracts these differences. FSC normalization should also be performed here to better assess whether this is a size effect.”

We thank the reviewer for the interest in the matter of the PRP boost, and for providing us with evidence that shows that others have seen this phenomenon (Shaw et al. 2019, PMID: 30955892), which we failed to notice at first. Beyond the graph in Shaw et al. 2019, we have since first submission found one other related observation: increased peptide production related to the PRP activation (Huberman et al. 2013, PMID: 24121774). We haven't been able to find more direct descriptions of this effect. The reviewer points out an interesting question about the nature of the PRP boost of yeast cells. As we understand, the two main hypotheses proposed are:

- i) Increased expression of antigen - i.e. an absolute increase of antigen, with an increase in antigen density on the cell surface.
- ii) Enlargement of the cell surface alone - i.e. an absolute increase of antigen, but without change in antigen density.

The product of the PRP boost results from an increase in the total amount of CD19 per cell, meaning that there must be an increased amount of expression per cell no matter the mechanism, which is evident from antibody-staining data (MFI) that is independent from cell size. Ste12+ yeast cells increase in size, so even maintenance of a constant antigen density on an increasing surface area demands an increase in expression. Our results indicate that expression is not limited by saturation, as the effect is equal for P_{PGK1} -CD19 and P_{TDH3} -CD19. So, we firmly believe that the antigen density increases, which is supported by the CAR T cell responses observed in this study, and furthermore supported by the comparison to differentially loaded microbeads, which shows similar responses to known increases in antigen density (**Fig. 3-4**). Most clearly, this can be seen when inspecting the difference in responses to P_{TDH3} -CD19 (constitutive promoter) for CAR TPR Jurkat cells; if we assume that surface area increases for P_{TDH3} -CD19 upon stimulation, but that the antigen density remains unchanged, then it should be the increased surface area, and hence absolute increase in antigen, that is the sole cause of increased stimulation of CAR T cells. Assuming that more yeast cells surrounding a single CAR T cell increase the engaged surface area of the CAR T cell, as indicated by our T/E ratio condition, we would then expect a similar activation response to increased GPCR-stimulation and hence surface area of the yeast cell. However, this is not the case - responses to an increased surface area (i.e. more target cells) differs from increased antigen density. At 8.0x P_{TDH3} -CD19 yeast cells per CAR T cell, there is still an increase in CAR activation, while for GPCR-stimulated 1.0x P_{TDH3} -CD19 yeast per CAR T cell, we observe overstimulation. As a positive control for antigen-density increase occurring in yeast, we used the PRP-orthogonal $P_{6xLexOLEU2}$ -CD19 strain that shows antigen-density increase, but does not engage other PRP-related genes, does not change its morphology (i.e. shmoo), and does not employ the PRP boost (**Fig. 4**). To avoid dependency on these more advanced phenotypes in yeast, future SCASA designs could be envisioned to employ such a system, as discussed.

Regarding the remark about SSC-/FSC-normalization, the short answer is that in our experience, SSC provides the best resolution of PRP-induced morphological changes. In fact, we had the same idea as the reviewer and we also performed FSC-normalization, which provided the same insights into the PRP boost effect outlined in the **“Additional logical analysis of the PRP boost effect”** (Suppl. Fig. 9). We observe that FSC and SSC are co-dependent parameters that are both seen to increase when investigating yeast cells undergoing PRP-induced morphological changes. However, the light-scattering observed in FSC correlates slightly less with mating pathway induction, compared to the light-scattering of SSC. Consequently, we have limited ourselves to interpreting these parameters combined as “morphological changes” or “cellular complexity”, as we have yet to differentiate them experimentally for yeast cells undergoing PRP-induction.

4. Points on the importance of cell antigen density vs cell ratios

“It is therefore unclear what is the key factor for CAR T stimulation, CD19 density per cell or total CD19 in the co-culture? This is crucial for understanding the best way for other researchers to take this system and apply it to CAR T cell activation. Can I just pick a single constitutive promoter and vary the density with cell count, as shown in Fig. 5?”

If so, is the inducibility aspect of this work necessary? It seems all of the data is here to better convey this message, but it needs to be presented in a clear manner. Additionally, the pros and cons of the system need to be discussed in far more detail.”

We thank the reviewer for raising this important question on the effects of CD19 density antigen density per cell and the total CD19 in the co-culture. We hypothesized that these parameters would impact CAR designs and downstream signaling differentially. So to answer this question, we designed the CAR TPR Jurkat experiment to contain both GPCR stimulation to increase antigen density and multiple target-to-effector (T/E) cell ratios (**Fig. 4**). We found that these are individual parameters for CAR T cell activation, with different dependencies for CAR designs and transcription factors, aligning with current CAR T cell activation research. Clear examples are responses to P_{MFAZ} -CD19 (e.g. NFAT-eGFP) and P_{TDH3} -CD19 (e.g. NF- κ B-CFP) (**Fig. 4b, Suppl. Fig. 19**). However, as the reviewer may also suggest in the above comment, it is clear that these parameters are connected, as a certain antigen density will only have an impact if there is a sufficient target-to-effector (T/E) cell ratio, and vice versa. Nevertheless, they cannot directly compensate for each other as CAR T cells are indeed sensitive to antigen densities, which leads to different responses, and we hope that this important point is clear from the new experiment (**Fig. 4**) and revised paper.

5. Combining Figure 2 and Figure 3

“Characterizing pheromone responsive promoters is fairly well-established and thus does not require a dedicated figure. Perhaps combining Figs 2+3 to show CD19 display can be tuned using PRP promoters – this is the novel part of the work. Inverse promoters are also not relevant to this work.”

We agree with the reviewer and think this is a great idea. We have combined the figures and result sections to adapt accordingly.

6. The matter of galactose-induction and P_{GAL1} in yeast surface display

“The $Pgal1$ system was optimized by integrating the construct into the genome. The rationale for not using this commonly used inducible promoter was a digital response. However, it is never contrasted with the GPCR system, only a constitutive promoter. Furthermore, $Pgal1$ can be converted into an analogue response with a single deletion of $GAL2$, which is not provided as a simple solution to the binary behavior. Arguably this would be far simpler than the multiple gene KOs required for using the PRP. Again, the motivation for not using a standard inducible promoter (or constitutive promoter) here is weak. Additionally, the improved integrated galactose inducible system should be the comparison in Fig3b, not the conventional EBV100 plasmid-based system, as this is instead a comparison of integration vs episomal, which is not the argument being made.”

We thank the reviewer for this comment. While we verify that genomically-integrated P_{GAL1} increases CD19 display

efficiency compared to standard plasmid-based expression in EBY100 (Suppl. Fig. 2), galactose induction was not considered a starting point for SCASA for the following reasons:

- Limited to a single possible input: galactose.
- Requires yeast growth in the absence of glucose - and hence a metabolic growth shift, and hence the system is potentially inhibited in glucose-rich mammalian media for co-cultivation studies.
- Has slow induction compared to other systems (days rather than hours), which we wish to minimize for applications within human immune cell-cell interactions that occur on a very short time-scale (*e.g. see pt. 15*).
- Offers limited customizability, despite successful efforts to convert the binary response dynamics into tunable responses, as the reviewer also makes us aware of (*GAL2 engineering*).

Rather than adapting an existing system for YSD, we aimed to explore diverse tuning knobs of cross-kingdom-conserved GPCR signaling in the context of YSD, and to demonstrate a novel application in cell-cell communication benchmarked against state-of-the-art methods, in the case of CAR T cell activation.

Furthermore, based on the feedback on comparisons, we have removed the comparison of SCASA to NALM6 and EBY100 + pCT from the main figure. Instead, we have added the NALM6 as a benchmark to the comparison of baseline CD19 levels, and directly compared it to P_{TDH3}-CD19, as this is a more relevant comparison in this new result section structure (i.e. all different cells are at baseline constitutive levels) (**Fig. 2g**)

7. Outlining pros and cons of SCASA versus other platforms

“In terms of the SCASA system being applied to study CAR activation, I recommend bolstering the discussion/rationale for why this approach may provide unique benefits vs. conventional NALM (or other cancer) cell lines engineered with different antigen levels. This is partly addressed in the observations in Figure 5 and in the discussion; 1) there is “CAR-dependent NALM6 antigen-down modulation” and 2) “cytotoxic sensitivity of NALM6 imposing a selective pressure bias and loss of CD19+ cells”. However, from a translational point of view, it still may be preferable to work with human cancer cells when evaluating different CAR designs to connect design, antigen density and desired phenotype (i.e. cell death which cannot be assessed with yeast). If a benefit of the SCASA system is to enable a more robust and high-throughput platform than can be achieved when using human cancer cells, this should be demonstrated or, at the least, further explained.

We agree with the reviewer, and in our vastly updated manuscript (incl. new benchmarking results) we have implemented detailed descriptions of the system features and dynamics compared to state-of-the-art antigen-density model systems, which is important both in the case of mammalian cell lines and non-cellular systems. As the reviewer also outlines, we conclude that some specific applications could benefit from using a robust and orthogonal platform, such as non-cellular platforms or yeast cells, while other applications demand less orthogonal systems,

such as cancer cells for cytotoxicity assays. Importantly, we have introduced new experiments that directly compare yeast performance to commonly used non-cellular material-based platforms (CD19-coated microbeads and microtiter plates) on several different parameters (e.g. activation, activation per CD19, potency, CAR response dynamic range, cost, and cost-effectiveness) within applications of characterizing CAR designs and downstream signaling (**Fig. 4c, Suppl. Fig. 20-24**) - please see pt. 18 for the specifications of these results. This is included in the result section “*SCASA yeast applications in characterizing CAR designs and downstream signaling*”. We highlight the experimentally determined differences of the yeast system in the results section, comparing it to non-cellular platforms in the new Fig. 4 (**Fig. 4c, Suppl. Fig. 20–24**) and mammalian systems in the revised Fig. 5, focusing on aspects such as system robustness across cancer cell lines to enable a direct comparison between NALM6 and SCASA yeast (**Fig. 5i**). Summary of comparison to cancer cell line:

- Yeast has a more robust antigen density than cancer cells (26.5-fold), which lost 98.5% of their surface antigen within 20 hrs. (**Fig. 5d-e,i**).
- Yeast provides more robust target cell numbers than cancer cells (17.0-fold) (**Fig. 5f,i**).
- Yeast and cancer cells could derive the same main conclusions on the functionality of the CAR T cell product (e.g. activation, CAR expression, CAR T cell retention), but yeast could more efficiently activate the CAR T cells, is more orthogonal, and provides more stable assay conditions (**Fig. 5**).

The benchmarking is done within the scope of the two different applications outlined in the result sections - i.e. systematic testing of CAR designs (Fig. 4) and batch testing of a clinically derived CAR T cell product (Fig. 5). In addition, we discuss the advantages and limitations of yeast-based, non-cellular, and mammalian cell line-based systems, which can be found in a paragraph of the Introduction and the Discussion. As for high-throughputness, the ease of handling, engineering, and growing yeast, as well as low cost, make it ideally suited for future parallelized screens of CAR designs or donor samples, as also highlighted in our new discussion and showcased in the new Fig. 4 (*see also pt.8 below*). We thank you for the suggestion and hope that the reviewer also finds the new experiments and perspectives to have improved the manuscript.

8. Showcasing demonstration and applications of SCASA

While not strictly essential, it would certainly enhance the paper to test >1 CD19 densities with primary T cells (Figure 5), to demonstrate CAR activity with a variety of CD19 densities. Only one yeast CD19 density (PPGK1) was implemented in co-culture with primary cells while several were quantified with Jurkat cells (Figure 4). This is partially addressed in supp figure 11 with the addition of two promoters (MFA2 and TDH3), however I am unsure what alpha factor was used in these figures and am concerned that there is not a significant difference in CD25 expression between induced and uninduced yeast expressing CD19. Overall, I think there is a missed opportunity to use these demonstrations with Jurkat and primary T cells to showcase in what settings and/or applications SCASA is beneficial, superior, or powerfully complementary to using human cell lines.”

We appreciate the recommendation to demonstrate further applicability and advantage of the SCASA system and the recommendation to show that CARs produce different responses using SCASA yeast. This led us to design the new CAR TPR Jurkat experiment with 50 different yeast-based stimulatory conditions (**Fig. 4**), resulting in further demonstration of the complementary use of SCASA yeast in investigating CAR T cell designs. Specifically, with comparison of characteristics for FMC63-CD8 α -4-1BB-CD3 ζ (tisagenlecleucel) (4-1BB CAR) and FMC63-CD28-CD28-CD3 ζ (axicabtagene ciloleucel and brexucabtagene autoleucel) (CD28 CAR). This experiment shows that yeast can be used in a high-throughput manner to interrogate CAR T cells and arrive at conclusions that corroborate the current knowledge of the field. We also demonstrate how several different CD19 densities affect CAR T cell signaling, and compare this to target cell ratios, as also suggested by the reviewer. In summary, we now demonstrate two different main applications of the SCASA technology, namely characterization of CAR designs and verification of novel CAR T cell products - which we reflect on in the discussion.

In relation to Suppl. Fig. 11 (now Suppl. Fig. 25), we agree with the reviewer and this experiment is now shortly referred to in the main text. The new experiment demonstrates these aspects of versatility and applicability better.

9. The logic behind the choice of CAR designs

“It should be explained why these two CAR constructs (FMC63-CD8.-4-1BB-CD3 and Hu19-CD8.-CD28-CD3) were chosen and why there was a change when going from Jurkats to primary. This is important since it has been indicated in citation 38 and referred to in line 388 that ICD CD-28 outperforms 4-1BB in relatively low antigen densities. It would have been great if it could have been demonstrated using SCASA that these CARs produce different T cell activation levels when varying antigen density.”

Throughout this study, we implement three different CAR designs: FMC63-CD8 α -4-1BB-CD3 ζ (“4-1BB CAR”), FMC63-CD28-CD28-CD3 ζ (“CD28 CAR”), and Hu19-CD8 α -CD28-CD3 ζ . Specifically, the 4-1BB and CD28 CARs are chosen as they are the most commonly applied FDA-approved CARs and are known to differentiate in their responses in relation to antigen density. The Hu19-CD8 α -CD28-CD3 ζ CAR was chosen as this was a novel type of CAR T cell product associated with lower neurotoxicity to be tested from a novel manufacturing method by collaborators (Mohr et al. 2023, PMID: 36750230). We hope that the reason for our choice is now clear from the first paragraphs of “Controlled human immune cell activation by SCASA yeast cells”, “SCASA yeast applications in characterizing CAR designs and downstream signaling”, and “Characterizing a donor-derived CAR T cell product using SCASA yeast cells”. The above-mentioned result sections reflect different applications, for which different T-cell backgrounds are used; Jurkats are implemented as cell lines for CAR design characterization, while primary cells are used in the batch testing of a donor-derived CAR T cell product, as if it were to come from a patient.

We thank the reviewer for this comment, and we hope the reasons provided satisfy the reviewer’s comment.

10. Minor comments:

“If PRP is being stimulated AGA1 might be expressed at higher levels, which may also explain the boost in CD19 display. This was not discussed but could be an advantage of PRP induction for yeast display purposes?”

We agree with the reviewer on this point - and this discussion could even be expanded to comprise the GPCRs, all PRP pathway signaling components, and all factors involved in yeast surface display. As yeast synthetic biologists, we are very interested in knowing the exact mechanism behind this, however, we find that a discussion of the exact mechanisms lies just beyond the newly refined scope of the paper, to expand on the discussion above (*cf. pt. 3*). We hope that the reviewers agree with this point.

“Line 83: Please expand on what you mean by limited control of antigen densities.”

This sentence has now been removed, and we now discuss the control of antigen densities in various antigen-density model systems in the discussion and describe it in the introduction.

“Fig. 4b: what are the cell ratios being used here? Why were these 4 promoters chosen vs the 8 tested in 3E? Please make it more clear what the threshold level is for CAR activation - is it when a condition is significantly different than 0uM of alpha factor?”

Cell ratios: We have included a specification of the ratio in the legend and in the text; a target-to-effector cell ratio of 0.2x - i.e. 5 CAR T cells per yeast or NALM6 cell.

Promoters: These 4 promoters were chosen from the CD19 analysis, as they showed the highest absolute levels (P_{TDH3}), highest dynamic range (P_{FUS1}), highest span in absolute CD19 levels (P_{MFA2}), and provided a non-extreme constitutive CD19 level (P_{PGK1}) (**Fig. 2**).

Threshold levels for CAR activation: We have quite drastically reformatted this paragraph to more concisely explain what is meant, we hope that the reviewer finds it more clear as well (now **Fig. 3**). The main point is that changes in antigen-density are effectively detected for P_{TDH3} and P_{MFA2} , but not for P_{FUS1} , as the antigen levels are below threshold levels of responsiveness of the CAR at this low T/E-ratio. These dynamics are now thoroughly investigated in the new CAR TPR Jurkat experiment (**Fig. 4**).

Specifically, CAR activation is given as a significant difference from background (**Fig. 3c**) and increased CAR activation from GPCR stimulation is relative to the 0 μ M (**Fig. 3b**).

"Fig. 4c: How were %CAR+ Jurkat populations made?"

This was done by mixing a 100%CAR+ culture with a CAR- culture in ratios defined by cell counts, and is described in the methods section "CAR Jurkat NFAT-Luc activation assay with SCASA yeast cells and NALM6".

"Line 192: It is not surprising HPF1 and CSS1 are downregulated as the mRNA data already showed this. Again, inverse promoters are not relevant to this work."

We have removed this from the paper.

"Line 225: This is not sufficient evidence to say conventional YSD methods are unfit for high-precision cellular communication. Or if it is, you could comment on small improvements which could improve them, outside of the GPCR approach."

We agree that this statement was too decisive, as galactose-induction would at least enable binary cellular communication - however, in the revised paper we do not engage in this discussion, for the reasons we have previously outlined in relation to inducible systems.

"Line 378: What is meant by 100% CAR+ Jurkat NFAT-Luc cells. What control is this compared to?"

This means that 100% of the Jurkat NFAT-Luc cells are CAR+ - i.e. every cell in the culture expresses a CAR. This is compared to the unengineered Jurkat NFAT-Luc culture (0%CAR+), where none of the cells express a CAR. We hope that the updated descriptions of the experiment, the figure legend, and the methods make that clear now.

"Fig 5a: What concentration of alpha factor was used?"

No alpha-factor was used here, only different T/E ratios.

"Line 544: inverse agonism of PRP is not a novel insight (ref 60) and PRP-induced expression boost has previously been shown (ref 53)."

We have removed the aspect of inverse agonism, and have introduced a reference to the study that previously observed expression boost; Shaw et al. 2019 (PMID: 30955892).

“Line 566: Please expand on this: “Hence, we specifically regard the SCASA yeast platform as useful for applications where specific signals must be isolated from the complex interaction network”

We have dedicated a large proportion of the discussion to this exact matter, as per other reviewer comments, and we hope that this point has been clarified in the new discussion. The updated version of this sentence is now put in direct context of the alternative choices; mammalian cell lines or non-cellular platforms for antigen presentation.

Response to reviewer #2:

11. Dynamics of T cell/tumor cell interplay

“However, the system presented here is artificial and lacks the critical dynamics of T cell/tumor cell interplay.”

We thank the reviewer for raising awareness about the comparison of this artificial system in relation to the actual interaction between CAR T cells and cancer cells. We understand the reviewer's concern about differences in the dynamics of T cells and cancer cells. Hence, we have now dedicated the majority of the discussion to explore this exact topic. In brief, as we discuss, artificial antigen-density systems rely on reductionist designs to gain greater control of antigens and to eliminate confounding factors of cancer cells. As described above (*Reviewer 1, pt. 7*), we have strengthened the comparison between yeast and cancer cells in Fig. 5, and have introduced new experiments with comparisons to non-cellular platforms in Fig. 4. We hope that the new experiments and new formulation of the introduction and discussion sufficiently highlights the differences, advantages, and limitations of artificial systems compared to cell lines.

12. Differences in CD19 levels and CAR T cell activation

“There is a notable difference in the levels of CD19 between NALM6 and the yeast system, and the degree of T cell activation varies significantly across these models.”

We thank the reviewer for pointing out that the reasons for the differences between these models were not clearly explained. We hope that it is clear from the revised paper that this observation is an intentional part of the study, and a main goal of the study; i.e. that CAR T cell activation is dependent on different levels of CD19, and that this can be effectively shown using yeast. The degree of CAR T cell activation that significantly differs across these models occurs because of the confounding effects of the NALM6, as we have further highlighted in the new sections describing Fig. 5. The yeast was designed exactly to provide different antigen levels (Fig. 1,2). Importantly, we see in all examples of CAR T cells co-cultivation that the yeast system can achieve activation levels that are equal to NALM6, as well as lower and higher than NALM6, dependent on the intentional adjustment of the two main parameters: antigen density and target-to-effector cellular ratio (Fig. 3,4,5).

13. Differences of the yeast system

“As acknowledged by the authors, the yeast system presents considerable differences compared to standard cell lines. It's easy to engineer tumor cells with a gradient of antigens using sh-RNA, CRISPR-Cas9-KO, inducible systems, or viral vectors in a lab, while the yeast system might be challenging to implement.”

We thank the reviewer for bringing these concerns to our attention. As mentioned above, we have now introduced descriptions and discussions of other antigen-density model systems, and compared them to yeast. To briefly

comment on this, in our experience with yeast synthetic biology, human cell line engineering, and handling of primary T cells, we have experienced that it is remarkably easier, faster, and cheaper to work with yeast - albeit some chassis expertise is of course needed. We have highlighted these aspects with references to selected studies of yeast-based research within the immunology field, most of which we have additionally covered in a recent review: Deichmann et al. 2024, PMID: 38816333. We also now discuss that there are certain applications where mammalian cell lines can be more appropriate, while the cost-effectiveness of yeast can be a benefit for other applications. Lastly, SCASA toolkit cells will be made readily available upon manuscript acceptance for any mammalian cell line engineering lab and the broader research community, which will largely simplify the adoption of SCASA for CAR design and/or donor cell line screens. We hope that the updated discussion provides sufficient reflection on the advantages and disadvantages of both yeast and mammalian cell lines.

14. Natural targets in the yeast system

“Natural targets are not expressed in the yeast system, thus raising concerns about whether these findings can be generalized.”

We thank the reviewer for raising an interesting point. By natural targets, we understand molecules targeted by other receptors, such as accessory T cell receptors. We now describe, in the revised introduction and discussion, that both cellular and non-cellular antigen-density systems exist with different features, of which natural targets are one such differing feature (e.g. related to PRRs). As we now emphasize, many antigen-density systems in fact lack these and other features, however, we also see that studies have started to integrate natural targets in antigen-density systems, including yeast-based systems, as now discussed. These are interesting features that one could imagine would be interesting to integrate into an improved version of the SCASA yeast system. We hope that these new aspects sufficiently address the reviewer's concerns.

15. Long-term mechanistics

“Moreover, long-term mechanistic studies are lacking.”

We are happy to hear that the reviewer is interested in further characterization of the system. In relation to assessing CAR T cell activation, we think that a 24 hr. co-cultivation period is already on the long-term scale, with CAR signaling occurring on a second-minute scale (PMID: 30131370), and CAR T cell effector responses within hours (e.g. Hu et al. 2021, PMID: 34901832, and Majzner et al. 2020, PMID: 32193224). The co-cultivation periods were actually picked to satisfy the resolutions of the read-outs, for example for the CAR TPR Jurkat cell line (NF- κ B, NFAT, AP-1) this resolution was shown to be best at 24 hrs (Jutz et al. 2016, PMID: 26780292). We agree that even longer stimulation periods could be interesting in relation to understanding the impacts of antigen density on exhaustion, sustained effector responses, and differentiation. Additionally, characterizing the potential long-term effects of yeast could

provide valuable insights into the limitations or advantages of the SCASA system for long-term applications, although this lies beyond the scope of the present manuscript. We agree that studying long-term mechanistic focusing on the above aspects is an interesting follow-up study extending from our established platform herein.

Response to reviewer #3:

16. Use of yeast as an APC

“While the presented yeast system offers some degree of “diability” (please also refer to my comment below), it comes with a number of ambiguities related to the biology of the CAR-T-cell-APC interaction. (a) The authors are right about mentioning them in the discussion section, but underestimate the consequences resulting from co-incubation, some of which may not even been that well studied. I noticed that the co-culturing was limited to 20 hours, probably for good reasons, as yeast grows faster than T-cells, competes for nutrients and contaminates the media with yeast-related metabolites.”

We appreciate the reviewer’s very important comment on yeast growth and competition in co-cultures, as this was initially also a concern for us. However, our initial concern was based on assumptions - in fact, when tested we found no evidence for such problems, and as we have included in the revised manuscript, other studies that employ yeast for T cell co-cultivations have seemingly not encountered such problems either. We are confident in the ability of yeast to co-cultivate with T cells based on the following observations. Firstly, we have shown that yeast does not initiate growth in T cell media within at least the first 22 hrs., which is a long duration for *S. cerevisiae* that normally doubles within 3 hours, as also demonstrated for the controls in yeast media. We do not anticipate an onset of yeast growth at any point during further extended co-cultivations. The co-cultivation time points were not chosen because of any limitations with yeast, in fact they were chosen to provide the most optimal read-outs for CAR T cell activation (*please also see response to reviewer #2; pt. 15*). In addition, we have shown that the presence of yeast does not affect viability or proliferation of T cells isolated from a healthy donor for at least 96 hrs., even in the extreme condition where T cells are outnumbered 10-fold by yeast (**Suppl. Fig. 13**). Lastly, we do not see that the T cells express any response from the sole presence of yeast in relation to; CD69, CD25, NF- κ B, NFAT, AP-1, proliferation, viability, or CAR expression levels. These parameters are only affected when the yeast is CD19+ (**Fig. 3,4,5**).

“(b) There is a lot to be said about the geometry of membrane apposition, with the dimensions of the intermembrane space and the lengths of receptor-ligand pairs regarded as a critical factor for kinetic segregation of kinases and phosphatases (and hence CAR-proximal signaling). In this regard, the yeast system with the cell wall serving as anchor for CD19 (or other TAAs), is poorly defined. I think there are more defined and equally available/accessible platforms that are up to the task of ligand scalability (see (iii) below).”

We thank the reviewer for bringing this important matter to our attention. The CAR immunological synapse is indeed an interesting space, and in response to this comment, as well as feedback from the other reviewers, we have dedicated the discussion to unravel this matter. We hope that the reviewer will find it interesting. Interestingly, as should now be clear from the discussion, other non-cellular antigen platforms are also less defined, as these are mostly antigen fixed to a planar surface. Nevertheless, CAR synapses are studied with great success using such platforms (e.g. Gudipati et al. 2020; PMID: 32632291 and Burton et al. 2023; PMID: 36598945). However, bottom-up engineering enables approximation of these native physical properties, as shown for both yeast and plate-based systems (Burton et al. 2023; PMID: 36598945 and Smith et al. 2018; PMID: 29733631). We now also discuss such aspects. On another note, CAR T cell synapse formation differs from that of T cells (Xiong et al. 2023; PMID: 37715447), which likely enables these more simplified antigen model systems (Burton et al. 2023; PMID: 36598945). Additionally, CAR T cells are naturally challenged by cancer-cell-surface variations, such as membrane topography, cortex stiffness, and glycocalyx structures that they have to overcome (Xiong et al. 2023; PMID: 37715447). We believe that these aspects can partly shed light on how the CAR T cells can similarly respond to the yeast cells, as if they were cancer cells.

With the impressive engineerability of yeast, various parameters related to controlling interactions in this space can be explored in high-throughput. As such, the yeast cell wall anchoring of antigens should become more highly defined within the coming years, and so far has not posed an insurmountable challenge in studying other protein-protein interactions (Younger et al. 2017, PMID: 29087945).

17. Broad range of expression

“Figure 3D illustrates the ranges of scalable CD19 expression in various yeast systems. The dynamic range is limited and the range of surface expression is rather broad, also for the P_PGK1-CD19 variant, which is employed in experiments involving primary human CAR-T-cells. Such broad expression and overlapping expression levels (due to a limited range in expression) results in undesirable noise, which may render a detailed analysis of the heterogeneity in the response of any given CAR-T-cell population challenging if not impossible.”

We thank the reviewer for these concerns, as it has driven us to greatly improve the demonstration of the application of SCASA yeast cells. We will challenge these concerns, as our data now show that detailed analyses of CAR T cells are possible with respect to the yeast CD19 profiles: we demonstrate that our designs cover a dynamic range of 1724-fold CD19 levels (**Fig. 2**), and that these can induce differential CAR T cell activation, from undetected levels and up to 62-fold increase in NFAT activity (**Fig. 3**) and 59-fold increase in CD69 expression (**Fig. 5**), both above NALM6 cancer cells' levels of activation. We further demonstrate that NALM6 in fact shows more heterogeneous behavior than yeast in relation to antigen density (26.5-fold) and target cell numbers (17.0-fold) (**Fig. 5**), hence making yeast a more stable provider of signals over time. To further address this concern of the reviewer, and to address some points of other reviewers, we designed a large-scale experiment to compare the performance of

different CAR designs in relation to antigen densities and target-to-effector ratios (**Fig. 4**). We here demonstrated the high-throughput applicability of yeast to provide a detailed analysis of two different CAR designs, by setting up 50 different yeast-based stimulatory conditions.

We agree that the intermediate stimulation levels for some promoters display more heterogeneity, compared to some examples from the literature (Salzer et al. 2020, PMID: 32820173, Patel et al. 2024, PMID: 38177315, and Majzner et al. 2020, PMID: 32193224). However, as we have demonstrated (**Fig.3,4**), this heterogeneity does not prevent successful CAR T cell analysis. We agree that this is a design aspect of the yeast system that can be improved, which we believe can be resolved in future designs of the SCASA system, for example by potentially implementing a fully orthogonal GPCR-sensing system in a future study (e.g. Shaw et al. 2019, PMID: 30955892).

18. Other platforms than the yeast system

“There are other platforms available which allow for the titratable presentation of TAAs to CAR-T-cells.

(a) Electroporation of target cells with capped in vitro-transcribed mRNA (PMID: 32820173) gives rise to narrow ranges of expression (for each mRNA concentration, e.g. in Jurkat cells) with a highly dynamic range of expression (depending on the amount of mRNA added to the electroporation). Advantage: highly titratable with narrow expression.

(b) CombiCells (PMID: 38177315): The use of the Spy-catcher system allows for narrow and highly titratable surface display of recombinant antigens as well as accessory factors.

(c) (Strept-)Avidin-based solid platforms (e.g. beads or planar surfaces) allows for highly titratable and quantifiable surface display of recombinant antigen and accessory molecules. It is highly scalable and adequate for high-throughput analysis.

(d) Protein-functionalized planar glass-supported lipid bilayers (PMID: 32632291): advantages are as described in (c) but the system also allows for dialing in the lateral mobility of membrane associated recombinant antigens and accessory factors while it is still compatible with medium-throughput analysis.

All platforms are highly quantitative and support co-culture periods longer than 72 hours. (b) allows for killing experiments, while (c) and (d) still afford quantitation of degranulation (with the use of an anti-lamp-1 antibody).

Taken together, I did not perceive any advantages of the yeast-based system in the special context of cell-cell recognition that would make me choose it. Rather the opposite is true. Yeast is a fungus triggering TLR-responses in T-cells (as discussed by the authors) and with a cell anatomy that is considerably smaller than that of T-cells. This leaves an open question as to whether a given CAR-T-cell is associated with one, two or more yeast cells at any given time (e.g. at an E:T ratio of 1:5) further complicating the analysis.

In conclusion, I see interesting aspects of the presented study within the area of synthetic biology, yet I find it difficult to find additional value within the yeast system for the study of cell: cell interactions, as there are already more defined and highly accessible systems available to the community that support population / single cell as well as high throughput analyses.”

We thank the reviewer for the detailed thoughts and perspectives, which have helped sharpen our focus and broaden our awareness of these other intriguing systems. Based on the important points raised here and comments from other reviewers (e.g. *pt. 7*), we have conducted new experiments to benchmark yeast cells against other platforms: CD19-coated microbeads (5.5 μm) and CD19-coated planar surfaces (microtiter plates) (**Fig. 4, Suppl. Fig. 20-24**). This platform comparison is framed within an application case, where we evaluate yeast, microbeads, and microtiter plates for assessing antigen density effects on two CAR designs (4-1BB and CD28 CAR) in CAR TPR Jurkat cells. We focus on comparing microbeads and yeast in Fig. 4, as they are similar in size and shape and function as discrete units, enabling precise control of the target-to-effector ratio and determination of the number of CD19 molecules per target. In contrast, the microtiter plate lacks these features due to its continuous planar surface. To understand differences in both performance and practicality of the platforms, we evaluate different parameters that we summarize and reflect on here:

- **Activation per CD19 molecule:** Yeast provides a significantly higher activation per CD19 molecule per target than microbeads (**Fig. 4c, Suppl. Fig. 22**). The activation per CD19 molecule cannot be determined for the continuous microtiter plate surface. *Example:* the 28.9 ± 2.6 -fold higher activation per CD19 of P_{PGK1}-CD19 (917 CD19/cell) relative to microbead D (29,744 CD19/microbead) ($p < 0.0001$) averaged across all CAR TPR Jurkat parameters (**Suppl. Fig. 21, Suppl. Fig. 22**).
- **CD19 antigen density per target:** The employed microbeads can load more CD19 molecules per target than what is expressed by the unstimulated yeast designs introduced in this paper (**Suppl. Fig. 20, 24A**). However, the yeast cell antigen density range that convey differences in CAR activation (248 - 3,607 CD19/target), are close to the lower range of antigen levels found in some relapsing patient DLBCL cells compared to microbeads (10,802 - 81,901 CD19/target): before therapy, DLBCL cancer cells had 5,810 CD19/cell, which dropped to 2,021 CD19/cell in relapsed cells following CAR T cell therapy (axicabtagene ciloleucel) (Spiegel et al. 2024). The employed microtiter plate seemingly loads more than the microbeads.
- **Potency:** Yeast-based activation was 64.3 ± 14 -fold more potent than microbeads (**Fig. 4c, Suppl. Table 13**).
- **Activation:** Microbeads could induce a higher peak activation level compared to yeast, with the biggest difference being 1.59 ± 0.09 -fold between yeast and microbeads for the 4-1BB CAR NF- κ B response. Microtiter plates provided the highest maximum activation level (**Suppl. Fig. 21**).
- **CAR response dynamic range resolution:** Yeast and microbeads could equally well determine fold differences in NFAT, NF- κ B, and AP-1 responses; however, microtiter plates provided the highest resolution (**Suppl. Fig. 23**).
- **Cost of reagents:** yeast was 1,005-fold cheaper than microbead-based experiments, and 878-fold cheaper than microtiter plate-based experiments (**Suppl. Table 9**).
- **Activation per cost and cost-efficiency (activation per CD19 molecule per cost):** yeast provided remarkably higher activation per cost than both microbeads and microtiter plates (**Suppl. Fig. 24B-D**). Yeast provided a

higher cost-efficiency than microbeads (**Suppl. Fig. 24C**) - example: the similar activation of P_{T_{DH3}}-CD19 yeast and 4 µg/mL (C) CD19 microbeads was 20,004±1,938-fold more cost-efficiently achieved by yeast at 0.1% of the cost.

Despite the possibility of a TLR response elicited by yeast, we observe no effect by simply co-cultivating yeast cells with human immune cells, please see above; *cf. pt. 16*. Beyond the current study, omics-based approaches will serve to further characterize the orthogonality of yeast to the human immune system.

The cell size of yeast is highly similar to that of T cells (5-10 µm). We have also addressed a related concern in great detail, antigen density versus target-to-effector ratio; *cf. pt. 4*.

In addition to the novel experiments, we have implemented nuanced discussions of both mammalian cell lines and non-cellular systems in the revised Introduction and Discussion - including systems that the reviewer has highlighted here. We have dedicated much space to these aspects in the revised manuscript, and now discuss the advantages, disadvantages, and practicalities of yeast and other technologies, as we also find this matter important. In addition to benchmarking against non-cellular platforms, we have strengthened the comparison of SCASA yeast cells to the commonly used NALM6 cells throughout the study (**Figs. 2,3,4,5**), including a stronger and more focused assessment of their compared performance as target cells (**Fig. 5**).

In the perspective of existing platforms, our ambition is not for SCASA to currently outperform established methods, but to explore complementary new ways to innovate research in human therapies. On this note, we find that SCASA expands the field of immunology to new pools of researchers with different ideas and backgrounds. In summary, we show that yeast offers a cost-effective microbial alternative or complementary approach to non-cellular antigen-presentation platforms. As noted in the discussion, yeast would benefit from further optimization, for example, to display a wider range of CD19 densities and to reach activation levels comparable to those achieved with microbeads and microtiter plates. To demonstrate the versatility of current yeast designs, we performed large-scale characterization of 4-1BB and CD28 CARs using yeast alone, applying 50 distinct stimulatory conditions to probe various target-to-effector ratios and antigen densities via GPCR-mediated stimulation (**Fig. 4**).

We thank the reviewer for the valuable suggestion and hope that the new experimental results, discussions, and perspectives have helped improve the manuscript and better contextualize the yeast platform among alternative approaches.

Other corrections:

- In the methods section "*CD19 display characterization*", we have added an extra antibody used for staining of HA-tags.
- In the methods section "*Yeast transformation and engineering*", we have added a description of the building of strain DIX67.
- In the methods section "*Human cell engineering and cultivation*", we have added a description of how the CD28 CAR and 4-1BB CAR TPR Jurkat cells were manufactured.
- In methods, we have added specific yeast strain names for the "Processing module promoter characterization", "CD19 display characterization" and CAR Jurkat NFAT-Luc experiment.
- In methods, we have added the new section "*CAR design screening assay in TPR Jurkat cells using SCASA yeast cells*" to describe the methods of the new experiment that was requested by the reviewers.
- In the methods section, we have added: "*Comparison of SCASA yeast cells to microbeads and microtiter plates*".
- In methods, we have added the detail that CTRL T cells were also transfected, but without CAR HDR template ("*Isolation of T cells from peripheral blood and CRISPR-MAD7 engineering for CAR insertion*")
- In figure legends, the significance levels have been defined correctly - ns: $P > 0.05$, * $P \leq 0.05$, ** $P \leq 0.001$.